# Assessing floods and droughts in the Mékrou River Basin (West Africa): A combined household survey and climatic trends analysis approach

Vasileios Markantonis[1], Fabio Farinosi[1], Celine Dondeynaz[1], Iban Ameztoy[1], Marco Pastori[1], Luca Marletta[1], Abdou Ali[2], Cesar Carmona Moreno[1]

[1] European Commission, DG Joint Research Centre, Ispra, Italy
[2] AgrHyMet, Niamey, Niger

*Correspondence to*: Vasileios Markantonis (vmarkantonis@gmail.com)

**Abstract.** The assessment of natural hazards such as floods and droughts is a complex issue demanding integrated approaches and high quality data. Especially in African developing countries, where information is limited, the assessment of floods and droughts, though an overarching issue influencing economic and social development, is even more challenging. This paper presents an integrated approach to assess crucial aspects of floods and droughts in the transboundary Mékrou River basin (a portion of the Niger basin in West Africa) combining climatic trends analysis and the findings of a household survey. The multi-variables trend analysis estimates at the biophysical level the climate variability and the occurrence of floods and droughts. These results are coupled with the analysis of household survey data that reveal behaviors and opinions of the local residents regarding the observed climate variability and occurrence of flood and drought events, household mitigation measures and impacts of floods and droughts. Based on survey data analysis the paper provides a per-household cost estimation of floods and droughts that emerged during a two years period (2014-2015). Furthermore, two econometric models are set up to identify the factors that influence the costs of floods and droughts of impacted households.

## 1. Introduction

Extreme meteorological events like droughts and floods represent an important limitation for the development of the poorest countries, impacting in particular the most vulnerable portion of the population. In these countries, agriculture remains the main economic activity and farming practices are mainly represented by rainfed agriculture (Rosegrant et al., 2002). Agriculture, a sector extremely vulnerable to extreme events, is the main source of income, or more often represents the self-sufficiency, of the poorest portions of the rural population in the least developed countries, as in the case of Sub-Saharan Africa (Gautam, 2006; Hellmuth et al., 2007). Extreme events cause loss of lives, damage to dwellings and vulnerable rural infrastructures, reduction of capital stock, agricultural and industrial production losses, and threaten, more in general, food security and development causing shocks in labour productivity, energy security, and political instability (Dell et al., 2014; Hsiang, 2010). Flood and droughts, jointly with the other natural disasters categories, were found to be particularly impacting in the African continent, in particular for what concerns fatalities, affected population, and economic damages (Cavallo, 2011).

A recent study (Shiferaw et al 2014), for instance, found that frequent drought conditions have limited the economic growth of many African countries and frustrated the benefits derived from development strategies implemented in other economic sectors, confirming also the findings of a previous study (Toya and Skidmore, 2007). Precipitation decline was found to be responsible of about 15 to 40 % of the gap between per capita Gross Domestic Product of the African economies respect to the rest of the developing world (Barrios et al., 2010). The negative impact on economic growth of increasing dry conditions and precipitation stresses was confirmed also by Berlemann and Wenzzel (2015) that found drought prone countries generally characterized by a lower education level, lower savings rate, and higher fertility. For these reasons, in the recent past, the disaster risk management community has extensively worked on the development of methodologies aimed at monitoring the risk prone areas and the overall vulnerability of the population threatened by the hydro-meteorological hazards. Progresses have been made in the assessment of the occurrence of extremes events, their magnitude, and the expected climate change impacts. Additional efforts were directed towards the improvement of both the assessment of risk and the estimation of the direct and indirect impacts, in particular related to loss of human lives, economic activities, infrastructures, natural and man-made capital. Technical advancement efforts allowed also an improved assessment of current mitigation measures and policies. The benefits derived from the progresses made in this discipline, however, were mainly concentrated in the most developed and technically advanced countries, where information is more easily available and mitigation strategies are more likely to be effectively implemented. In the case of the African countries, the assessment of the physical components of the hazard side followed the general technical development: as in the most advanced countries, in fact, the assessment of the occurrence of floods and drought has been conducted through the application of remote sensing based techniques analysing precipitation and temperature records and their spatio-temporal distribution, as for instance in the case of Ngigi et al. (2005) for a case study in Kenya.

Regarding cost estimation and impact assessment, instead, the knowledge about losses caused by past extreme events is still limited for a detailed quantitative analysis in many of the African countries. As described in Markantonis et al. (2012 and 2013), the methodologies used in literature are various: Hedonic Pricing; Travel Cost; Cost of Illness Approach; Replacement Cost; Contingent Valuation; Choice Modeling ; and Life Satisfaction Analysis (Welsch and Kühling, 2009; Luechinger and Raschky, 2009; Welsch, 2006). In this study, however, it was decided to estimate the cost of natural disasters in the case study area through the direct testimony provided by the affected population. Several studies made use of household surveys to acquire qualitative and quantitative information from the local population and use the knowledge collected to estimate the damages of past hydro-meteorological events (as for instance in: Fitchett et al., 2016; Ologunorisa and Adeyemo, 2005). Depending on the key economic sectors of the case study area, various examples of cost and impact assessment could be listed. Among the most representative, could be mentioned studies analysing the following sectors: health (Schmitt et al, 2016), agriculture and food security (Ngigi et al., 2005; Shiferaw et al., 2014; Shisanya and Mafongoya, 2016), and tourism (Fitchett et al., 2016). The analysis of the hazard components and the various impact assessments conducted in the recent past, allowed the evaluation of risk mitigation measures and adaptation strategies. Shisanya & Mafongoya (2016), for instance, analysed the effectiveness

of shifts in agricultural practices; Brower et al (2009) focused on the evaluation of the installation of risk mitigation infrastructures; Oyekale (2015) analysed the implementation of more advanced forecasting and early warning systems; Halsnæs &Trærup (2009) focused on the implementation of specific integrated policies and practices for flood and drought. In some of the mentioned studies, cost-benefit and willingness-to-pay analyses were used to add quantitative evidence to the qualitative description of the case studies in object.

The objective of this paper is to assess the occurrence of floods and droughts events as well as to estimate of damage costs at the household level and finally, the current mitigation behaviours adopted by the population in the Mékrou river basin, a small catchment in Western Africa. It combines a quantitative approach for detecting hydro-meteorological hazard prone areas (through the analysis of gridded climate datasets) with a quali-quantitative analysis of a household survey. This approach allows comparing physical analysis of extreme events with human perception of the flood and drought phenomena in the Mékrou river basin. The household survey collects sufficient information to conduct an estimation of the natural hazards impacts in terms of economic cost, and to present the most widely adopted mitigation behaviours.

Mékrou is a sub-basin of the Niger River, covering an area of 10,635 km², about 3% of the total Niger Basin surface, crossing the borders of three countries: Benin (80% of the basin territory), Burkina Faso (10%) and Niger (10%). Mean annual precipitation ranges from a maximum of about 1,300 mm in the southern region and 500 mm in the north, being the wet season between June and September with an average cumulated rainfall of 700 mm. Temperature is also highly variable in space and time. Warmest and coldest months are April and September, respectively. Mean annual temperatures spatially vary between 26-30˚C, being the maximum 35-40 ˚C and the minimum 15-19 ˚C. The Mékrou catchment is located in a temperate transitional area characterized by a wet season peaking in august and a long dry season spanning the period December-April (Masih et al., 2014). During the wet season, the whole Niger River basin is subject to regular floods and extensive research has been conducted on its complex hydrology made unique by the large system of lakes and wetland known as the Inland Delta (Bader et al., 2016; Tarpanelli et al., 2017), while lower attention was given to the Mékrou sub-basin. Flood and drought events are extremely frequent (Froidurot & Diedhou, 2017), while the impacts of climate change on their frequency and magnitude is unclear (Gautam, 2006). The Mékrou basin is characterized by lack or poor infrastructural development and very low socioeconomic conditions. Agriculture is the key economic sector with the arable land used for food production, cattle farming, and production of cotton. That for, climate variability constitutes the main threat for food and economic security of the area.

The paper structure is the following: Section 2 presents the methodological framework both regarding the development and application of the household survey (section 2.1) as well as the analysis of the biophysical variables (section 2.2). The findings of this integrated approach are shown in Section 3. This section analytically presents the findings regarding precipitation patterns, temperature and river discharge in the Mékrou river basin. Moreover, it includes the main findings of the household survey concerning observed occurrence of floods, droughts and climate variability, household mitigation measures, impacts

of floods and droughts and finally an econometric estimation of the costs of floods and droughts. Section 4 summarises the main findings of this approach, discusses its potential and limitations and presents the main conclusions.

## 2. Methodology

### 2.1 Households survey implementation and analysis techniques

A household survey aimed at evaluating several water related dynamics, including extreme natural hazards, was designed in 2015 and conducted in early 2016 (February to April). Specific villages and towns were selected to include a geographically representative sample of the river basin that belongs to the three countries (Benin, Burkina Faso, and Niger). The selection process was designed to keep a balance among urban and rural settlements. The number of the selected households proportionally represents the total population of each selected village or town as well as the number of households per country

represents the country's population within the basin. Household were selected randomly. . The survey was implemented by experts of the Joint Research Centre of the European Commission in cooperation with local universities from Benin, Niger and Burkina Faso. The Interviews were conducted in person by a team of students supervised by a professor for each of the countries. The students received a training before starting the survey where the questionnaire was thoroughly explained and discussed with them. Since the area is francophone, all the material used was written in French. The fact that the survey was

conducted by students of local universities facilitated the communication with the local population, overcoming possible language or cultural barriers.. The information included in the household survey aims to retrieve opinions and observations based on personal judgement. A main section of this survey was to identify and assess the impacts and costs of flood and drought events, climate variability as well as household mitigation measures. Regarding the occurrence of floods and droughts in the study area, two time periods were selected: 10 years (2006-2015) and 2 years (2014-2015). The logic behind this

selection was to identify the most recent events (2 years), for which the local households could still have a fresh memory on the impacts suffered while affected. The second time period (10 years) offers a longer assessment of climate variability and occurrence of floods and droughts, which could potentially still be evaluated according to personal judgement. The whole questionnaire is included in Appendix A.

The survey process resulted to the collection of 660 randomly surveyed questionnaires retrieved from the areas of the three countries (Benin, Burkina Faso, and Niger) that are located in the Mékrou catchment (Figure 1). Specifically, 332 questionnaires were collected in 16 villages from the municipalities (called "Communes" in French) of Banikoara, Kouande, Kerou, in Benin. 148 questionnaires were collected in 6 villages from the Communes of Diagaga and Tansarga in Burkina Faso, and 180 questionnaires were collected in 8 villages from the Communes of Falmey and Tamou in Niger. The total

number of surveyed households offers a more than 95% significance in the statistical findings (400 questionnaires minimum required for 95% significance rate).

Figure 1. The Mékrou river basin and the household survey area

Following the data cleaning and validation of the survey, the information collected was processed through statistical analysis including all the parameters investigated. The survey responses were evaluated using descriptive statistics aggregating data both at river basin and at country level. In this way, the findings were analysed at the river basin scale illustrating, at the same time, the differences among the three countries.

Apart from the statistical analysis, an econometric estimation was applied to identify the specific parameters that are highly correlated to the costs of droughts and floods that occurred in the last two years (2014-2015). Hence, besides the cost estimation based on the sample mean, two econometric models were set up investigating the determinants of flood and drought costs following a cause-effect logic. Performing a thorough multi-variate regression among costs of floods and droughts as stated in the survey and other covariates, such as socioeconomic characteristics of the population, impacts, and mitigation measures, led to the construction of models that could be eventually used to explain the costs of extreme hydro-meteorological events. Several types of regression models were tested, both linear and logarithmic. Eventually, linear multivariate regression models were selected since they fitted the selected variables with a higher statistical performance. Moreover, in order to ensure coherence and readability of the models, only independent variables whose P-value was less than 0.05 were selected. One independent variable with a P-value higher than 0.05 was included due to its importance in the context of the analysis. Its value (0.068), however, is still below the least acceptable P-value (0.1). R-square and F-test values were estimated as well as correlation was tested among the independent variables in order to avoid bias in the model due to collinearity among the selected variables.

## 2.2 Analysis of biophysical variables: precipitation, temperature, and river discharge

Precipitation and temperature patterns and changes are the main drivers affecting local population perception about water availability, especially in an area where the main economic activity is based on rainfed agricultural production. In the frame of household survey interpretation, quantitative estimates of these events are needed to understand the underlying direct (rainfall and discharge) and indirect (heat-waves) factors that have an impact on the respondents answers. Analysis of rainfall events above and below the long-term average distribution in conjunction with inter and intra-annual analysis are useful to depict anomaly patterns and trends, thus, they contribute to better understand, for example, main drivers of meteorological droughts. This is particularly relevant in order to compare local population perception on climate variables with quantitative estimates. Characterization of events above and below the long-term rainfall average distribution in conjunction with intra-annual precipitation analysis were studied. To complement this, we analysed river discharge regimes, as precipitation anomalies could be translated into hydrological droughts, thus, generating water resource imbalance, groundwater level decrease, reservoir depletion, etc. (Liu et al., 2016). Secondly, considering the increasing number of heat-wave events occurred during the last decade in Africa (Ceccherini et. al 2017), we have studied its magnitude and spatio-temporal evolution in the

Mékrou Area of Influence to explain possible misperceptions that could arise in the surveys. The objective was to compare both precipitation and temperature stress with survey's results, offering empirical evidence of their concurrence, or eventually explaining the underlying causes of possible contrasting results.

### 2.2.1 Precipitation pattern analysis

Annual and seasonal rainfall descriptive statistics were analysed jointly with the results of a Seasonal Kendall test for monotonic trend (SK) applied over precipitation data derived from the Climate Hazards Group Infrared Precipitation with Station data v. 2.0 (hereafter CHIRPS). This database has a spatial resolution of 0.05º, corresponding to approximately 5 km (at the equator) and covers all longitudes in the latitude range 50°S-50°N spanning a time horizon included in the period 1981-present days (Funk et. al 2014). The SK test used is based on Hirsch and Slack (1984) and it was applied over two different time ranges: the entire time series (about 35 years), and over the last ten years, the period considered in the household survey. The test is a modified version of the previously SK test proposed by Hirsch et al. (1982), and attempts to reduce variables serial dependence, being seasonal precipitation data serially correlated. Both are based on the non-parametric Mann-Kendall test (Mann 1945, Kendall 1975, Warren and Gilbert 1987). Magnitude of trends are calculated by means of Sen´s Slope Estimator and Kendall tau as the rank correlation coefficient.

Additionally, in order to identify precipitation anomalies, we calculated the Standardized Precipitation Index (hereafter SPI) proposed by McKee et.al (1993; 1995). This index could be applied over different time scales, each providing information about the impact of a given anomaly on the availability of water resources (WMO 2012). In this study, the 3 months and 6 months SPI (hereafter SPI-3 and SPI-6) were calculated, associated respectively to meteorological and agricultural droughts. For the purpose of this paper, being the Mékrou communities' economy mainly based on rainfed agriculture, we focused on the precipitation anomalies over the wet season, presenting the SPI-3 for the period June-August (JJA), and the SPI-6 for the period included between April and September (AMJJAS). The results are given in units of standard deviation, indicating how far a given precipitation event is below (drier events are associated with a negative SD) or above (wetter event, positive SD) the long term normal distribution. SPI values could then be interpreted following a classification scheme where standard deviations are categorized into different classes, each associated to different levels of wet or dry anomaly. In this case we have divided the values following the seven categories proposed by Agnew (2000), where previous thresholds defined by McKee et al. (1993) were replaced by alternative classes (Figure S1). Finally, in order to get a general overview of the anomalies in the different administrative units, we calculated the area percentage affected by each class over the entire time series.

### 2.2.2 Heat-wave analysis

Russo et al. (2015) defined the Heat-wave Magnitude Index daily (hereafter HWMId), as the maximum magnitude of the heat-waves in a year, where a heat-wave is defined as a period equal or higher than three consecutive days with maximum temperature above a daily threshold calculated for a 30-year long reference period. The index is based on daily maximum temperatures taking the intensity and duration of the event into account. In our case, the source used to retrieve the maximum daily temperatures is the ERA-Interim reanalysis dataset (Berrisford et al., 2011; Dee et al., 2011) available from 1979 onwards, with an approximate spatial resolution of 80 km at the equator. The index was applied to study these events for the last 35 years across the study area.

### 2.2.3 Discharge in the Mékrou river

In order to have a quantitative estimation of water availability within the Mékrou river basin, the hydrological model SWAT (Neitsch et al., 2011) was setup and calibrated to assess annual and monthly river discharge. The SWAT model integrates all relevant eco-hydrological processes including water flow, surface runoff, percolation, lateral flow, groundwater flow, evapotranspiration, transmission losses, nutrient transport and turn-over, vegetation growth, land use, and water management. SWAT subbasins were delineated using the ArcSWAT interface with a Digital Elevation Model with a 90 m spatial resolution, resulting in 32 subbasin for the whole area.

The historical discharge data recorded at the Barou gauge station (outlet of the river basin) and at the Kompoungou (draining area is about 56% of the river basin) gauge stations were used for model calibration. Lack of data availability represents a huge limitation for the hydrological analysis of the basin. The mentioned discharge observations, in fact, cover a limited and not overlying period (1990-2000 for Barou and 2004-2013 for Kompoungou) and are not complete. We used SWAT-CUP program and manual setup to calibrate outflow in the two monitoring stations reaching satisfactory efficiency statistics (Moriasi et al., 2007) at monthly scale. In Barou NSE (Nash–Sutcliffe model efficiency coefficient, (Nash and Sutcliffe, 1970) is 0.87 and linear regression R2 is 0.88 and in Kompoungou NSE is 0.77 and R2 is 0.71. We used the SWAT modeled discharges to consider an extended time period, required in particular to take into account climate variability, and to cover areas of the basin where observations are not available.

## 3. Results

### 3.1 Biophysical variables: precipitation, temperature, and river discharge

#### 3.1.1. Precipitation and heat-wave patterns

Mean Annual Precipitation in the study area varies from 500 mm in the northern administrative units (Bottou, Tamou, Kirtachi and Falmey), to about a maximum of 1,000-1,300 mm upstream in the southern portion of the basin (Kouandé, Kérou and Banikoara (Figure S2). Generally, rainfall is highly variable in time and space, and follows a cyclical trend of wet and dry periods. The mean monthly precipitation for the wettest month (August) in the southern and northern regions varies between 200 mm and 300 mm, respectively, with the driest months (November-February) being close to zero.

In the period under consideration, mean annual temperature ranged between about 30º C in the North to 26.5º C in the South. April was the warmest month, with a maximum temperature of 40º C in the North and 35º C in the South. Minimum temperature in the coldest month (September) ranged between 19º C in the North and 15º C in the South, presenting the semi-arid region the highest range of variation of about 25ºC. Comparing the precipitation patterns over a short- (2007-2016) versus a long-term (1981-2006), it was possible to identify a slight decrease of rainfall during August and September, while a moderate increase was detected during June and July (Figure 2). Long and short term Seasonal Kendall analysis, performed at a significance level α = 5%, however, was not able to identify any significant trend (Figure 2). Some spatial differences were identified analysing the last 10 years: in that regards, negative slopes were detected in the southern regions of Benin.

Figure 2. Comparison of seasonal distribution of precipitation for the first 25 years of the time series versus the last 10 years (left) and Seasonal Mann-Kendall results (right).

#### 3.1.2 Precipitation stress analysis

The results of the SPI-3 and SPI-6 analysis highlighted a period of moderate to extreme droughts during the earlier part of the 1980s; this could be also noticed in mean monthly precipitation values, where the overall rainfall amount from 1981 to 1985 is noticeably below the long-term average (Figure 3). After this period the positive or negative anomalies were found to be more erratic, presenting an alternated series of positive and negative events, such as some severe wet anomalies in the southern regions in 2003 or drought events in the north regions in 1997. Regarding last ten years, there are few drought anomalies affecting at least 40% of the administrative areas. In particular, the northern portion of the basin in Niger (communes of Tamou and Kirtachi) was affected by a moderate to severe drought event during 2011, while in 2014 a similar event hit the southern portion of the basin in Benin (Banikoara, Kérou, Pehúnco and Kouandé). On the other hand, severe to extremely wet anomalies, were recorded predominantly during the years 2003, 2005, and 2007 in the entire Mékrou Area of Influence. Summarizing the results derived from the analysis of the SPI, we found that the decade considered in the household survey was mainly

characterized by precipitation in line with the long-term average. The anomalies were not particularly intense and most frequently related to wet conditions if considering the meteorological precipitation stress indicator (SPI-3 - 18 wet versus 15 dry anomalies), while a predominance of dry conditions was recorded when considering the agricultural precipitation stress (SPI-6 17 dry versus 14 wet anomalies) (Figures S3 and S4).

Figure 3. Temporal monthly precipitation profile for the Mékrou Area of Interest

### 3.1.3 Heat-waves

The spatio-temporal evolution of the heat-wave magnitude index HWMI between 1981 and 2015 is presented in Figure 4. Despite some isolated events during 1987-1988 and some other extreme in 1998, the analysis highlighted a constant increasing trend of the heat-waves' magnitude starting in 2004. The spatial pattern, instead, is not clear, being the whole Mékrou Area of Influence affected. In 2005 the HWMI was found to be higher in the central and northern part of the basin, while in 2006 the highest values were recorded in the southern part, and in the northern in 2010 (Figures S5 and S6).

Figure 4. HWMI index computed for 1981 to 2015 on the Mékrou

### 3.1.4 River water flow trends

The annual river discharge resulting from model simulation for the period 1995-2012 is presented in Figure 5. The first 5 years of the simulation period (1990-1994) were discarded to consider an adequate model spin-up period.

20 Figure 5. Mékrou river daily average discharge at Barou station as modelled in SWAT in the period 1995-2012. Dashed line indicates the average of the total period under consideration.

Modelled discharge in the period under consideration averaged around 24 $m^3$/sec, corresponding to an average annual water flow of about 760 $Mm^3$ (ranging from 190 to 1,400 $Mm^3$ respectively in 1997 and 2008). Spatial distribution of the water resources follow the topography of the basin, making the headwaters, where annual discharge remains below 100 $Mm^3$, particularly subject to inter- and intra-annual variability (Figure S8). High annual variability of the river discharge is an important issue for the sustainable water use in the basin, especially considering that agriculture is the main economic activity and the lack of water storage infrastructures in the basin. Intra-annual discharge variability in the Mékrou river basin follows the precipitation seasonal patterns: most abundant flows are reached after the rainy season period (July-November), being the peak flow reached in the period Aug-Sep-Oct (Figure S7).

### 3.1.5 Analysis of the population perception on the occurrence of extremes events and climate variability

The household representatives stated their personal opinion on the occurrence of specific extremes events in the Mékrou basin during the last ten years (2006-2015) (Table 1). The selection of this 10 years framework allows and relatively mid-term assessment of the past climatic events. Regarding the occurrence of droughts, 86.8% of the households declared an increasing trend in the period under consideration. This percentage is lower in Niger but still considerably high (71.5%). In addition, the vast majority of the local population (88.5%) estimates that the levels of rainfall decreased during the last ten years. This result is particularly evident in Benin and Burkina Faso (93.7% and 95.3% respectively). Not only the rainfall decreased but also, according to the local population, (92.7%) the seasonal distribution changed. In fact, the majority of the respondents (83%) states that the rainy season started with a delayed onset during the last 10 years, whereas almost the totality (91.8%) declared an earlier end of the season. Regarding heat-waves, 76% of Mékrou's interviewed population (62% in Benin) stated that periods of intense heat-wave became longer during the last 10 years. Regarding the number of events experienced, an increasing number of drought events were recorded and appeared in frequency mainly from none to 6 events. On average, local population experienced approximately 4 drought events if considered the whole Mékrou basin (Table 2). The number is higher in Burkina Faso (4.8 events) and slightly lower in Niger (3.4 events).

Concerning flood events, the opinions are less homogeneous over the whole Mékrou area. The majority (62%) of the respondents stated that floods frequency did not increase over the past 10 years. This trend is quite heterogeneous among the three countries: 84.9% of the households in Benin agreed with the majority, the responses are discordant in Burkina Faso, whereas in Niger 72.2% of the respondents found an increasing flood frequency in the period under consideration. When asked about the numbers of flood events experienced, almost one third of the families stated that no flood events occurred. The remaining replies are mostly distributed among 1 (23.3%), 2 (17.4%) and 3 (13.1%) events. As for the change of the events frequency, this occurrence is different when looking across the three countries. In Beninese part, 60.7% of the population did not experience any flood during the last 10 years. On the other hand, the larger flood event numbers are recorded primarily in Niger, followed by Burkina Faso (Table 2). In Benin, only an average of 0.7 flood events was declared during the last 10 years, 1.8 in Burkina Faso and 3.5 in Niger. The combination of these results suggests that the Nigerien portion of the Mékrou basin was the most prone to flood events in the period under consideration. Regarding both floods and droughts the perceptions do not differentiate based on different socioeconomic characteristics apart from the heterogeneity among the three countries in the basin. This could be explained by the homogeneity of the socioeconomic characteristics within the basin although it is spread across three countries.

*Table 1. Observed climatic changes during the last 10 years: 2006-2015*

*Table 2. Statistical Analysis of reported number of floods and droughts during the last 10 years*

## 3.2 Floods and Droughts Household Mitigation Measures

This section provides an empirical analysis of the household mitigation measures. The respondent had the choice among 13 mitigations strategies aimed at coping with changes in temperature and rainfall patterns. 8 out of the original 13 strategies were considered negligible due to a positive response rate lower than 10%, including: practicing off-season agriculture; application of more intensive irrigation; raising less livestock in order to increase crops; raising less small ruminants for switching to more cattle; raising less cattle and switching to camels; raising less sheep for switching to goats; adoption of specific techniques to regenerate the necessary grass cover for the livestock; and rent or mortgage land.

The mitigation measures that recorded a positive a response rate above 10% are: change of crop seeds; terracing the soil or using other methods to protect against erosion; plantation of more trees; emigration of at least one household member; practicing more often non-agricultural activities as sources of revenue (Table 3). Changing crop seeds is a quite common strategy mainly to adapt to rainfall changes and is widely applied in Burkinabe and Nigerien areas (for around 60-65% of respondents), but also in Benin in a smaller extent (34%). The practice of terracing the soil for preventing erosion is mostly applied in the Beninese part of the Mékrou river basin. The plantation of more trees is a common adaptation practice across the basin, due to both changes in precipitation (31%) and changes in temperature (20%). Emigration appears as an adaptation strategy to mitigate the economic losses resulting from the impacts of changes in rainfall patterns. Around 25% of households saw one of its member migrating due to temperature and rainfall changes. Secondary, 16.4% of Beninese respondents state that at least one member migrated in the last 10 years due to rainfall changes. Finally, an important number of households, primary in Benin ( around 35%) and secondary in Niger (around 15%), are practicing more often non-agricultural activities as source of revenue to cope with the loss of income due to both rainfall and temperature changes.

*Table 3. Household mitigation measures*

## 3.3 Floods and Droughts impacts and their cost-assessment

The local population of Mékrou has additionally indicated the occurrence of extreme events during a two years period 2014-2015. The relative recent occurrence of these events, some of them still ongoing, allows a more thoughtful estimation of the impacts and associated costs. Only a small percentage of the respondents (68 out of in total 660 interviewed households) stated that they had not experienced any flood occurrence during the last two years (Table 4). However the occurrence of events differentiates among the three countries, whereas most of the events are reported in Burkina Faso (34.5% of the households affected by floods). Aligning with the frequency of reported events, the impacts of the floods are proportionally more common in Burkina Faso. The most commonly reported impacts regard the losses of agricultural production, damages to the houses and loss of livestock (Table 5).

Similarly, the same analysis was applied for the recent drought events. The vast majority (76.7%) stated that they have not experienced any drought event during the last two years. However, more droughts are reported compared to floods resulting that 152 of in total 660 interviewed households experienced an extreme drought event. Likewise, the occurrence of droughts are heterogeneous across the three countries. Droughts are less prominent in Niger (Table 4), where only a few droughts are reported, while in Benin one fourth of the population has experienced droughts during the last two years. In Burkina Faso almost half of the households have experienced droughts during the last two years. The impacts of the recent drought are mostly recorded in Burkina Faso and Benin (Table 5) presenting a different profile. In Burkina Faso the impacts refer exclusively to losses of agricultural production, while in Benin still the majority indicates the loss of agricultural production but states also malnutrition and loss of livestock as result of the recent droughts. Regarding impacts, for both floods and droughts the collected information relates to general categories of impacts, mainly agriculture, livestock, housing without being feasible to collect more in depth qualitative characteristics of the impacts.

*Table 4. Experienced extreme flood events during the last two years (2014-2015)*

*Table 5. Impacts of extreme floods and droughts to the households*

The cost of the recent floods is higher in Burkina Faso (Table 6), where the average cost for an affected household in 334,326 FCFA (West Africa Francs) (approximately 495 Euro in 2017). The estimated flood costs differ among the three countries whereas it is much higher in Burkina Faso (334,326 FCFA) than Benin and Niger (40,000 and 160,000 FCFA respectively). Furthermore, we observe a difficulty from the households to estimate the costs of the recent floods, especially in Benin and Niger, the less affected countries, where the majority of the affected households were not able to provide a cost estimation. 20 out of 68 in total affected households in the basin were not able to provide a cost estimation of the experienced floods.

The average cost of the recent droughts, 256,440 FCFA (~391 Euro), is almost the same in Burkina Faso and in Benin (Table 6) lower than the average cost of the recent floods. Again, in this case a difficulty is evident from the households to estimate the costs of the recent droughts, especially in Benin where the majority of the affected households were not able to provide a cost estimation. In this case, 61 out of the 152 households that were affected by droughts were not able to provide a cost estimation. Regarding the geographical differentiation of both floods and droughts costs the heterogeneity is observed among the three countries but not within the countries where costs are homogenous among the selected villages and towns. Additionally the small sample of the affected households by floods and droughts statistically does not allow the aggregation of the estimated values to the total population of the basin.

*Table 6. Estimated costs of the recent flood and droughts (in FCFA)*

### 3.4 Cost assessment of floods and droughts: an econometric estimation

Additionally to the statistical estimation of the floods and droughts costs, two econometric models were developed to provide a reasoning of costs with socioeconomic and other relative factors. A wide series of independent variables of the household, such as socioeconomic conditions and mitigation measures e.g. have been used to find the determinants of the extreme events costs and estimate the costs of floods and droughts. Using several regression models and combination of independent variables two models were set up including exclusively statistically significant independent variables (P-value less than 0.05 and in one case slightly higher than 0.05 but less than 0.1). Table 7 presents the total of the independent variables including their scaling and their correlation. The latter is important for the coherence of the multivariate models, since low correlation among the independent variables excludes the existence of multicollinearity.

*Table 7. Selected independent variables for modelling costs of floods and droughts*

Regarding floods, the multivariate regression model Table 8 includes as independent variable the self-stated economic status (ECONSTAT) of the households as a qualitative alternative to the household income. The model reveals a strong economic status effect meaning that the richer the households are, the higher are the economic impacts of flood. The negative sign is due to the structure of the variable scaling (1 meaning rich and 5 much worse economic condition than the other households). Additionally, two of the main flood impacts, loss of crop productivity (CropProdLoss) and loss of livestock (LivestockLoss) were included in the model as independent variables and were found significant. According to the multivariate regression model, the average cost of floods per household during the flood events of the last two years (2014-2015) was equal to 390.92 euro. In order to avoid problems of multicollinearity, correlation among the independent variables was tested. The calculated Pearson's R values, all below the 0.3 threshold, suggest a low correlation among the independent variables.

*Table 8. Costs of floods – Multivariate linear regression model*

Similarly, a multivariate regression model was applied to estimate the costs of droughts (Table 9). The independent variable related to loss of livestock was found to have a strong effect in this model too. However, drought costs are found to significantly depend on the total crop production of the households (PRODCROP). According to this regression model, the estimated cost of droughts per household that experienced drought events during the last two years (2014-2015) was 494.76 euro. Similarly low correlation was detected among the independent variables.

*Table 9. Costs of droughts - Multivariate linear regression model*

**4 Discussion and Conclusions**

This paper combines the results of a household survey and climate data analysis to assess floods and droughts as well as climate variability in the Mékrou river basin in West Africa. The opinions and perceptions of household representatives revealed a strong climate variability at a ten years period (2006-2015). It is worth mentioning that 83% of the population, during this period, noticed a delayed onset of the rainy season. In addition, 91% of the population observed also an anticipated end of the wet season. Moreover, 88.5% of the respondents reported a general reduction of the precipitation during the ten years period under consideration, and 75.9% reported an increase in magnitude and frequency of the extreme heat events. This tendency is partially confirmed by the analysis of the climatic variables, mainly based on precipitation and temperature data. The findings of the analysis confirmed the increase of both frequency and magnitude of the heat-waves in the area of study. The climatic variability was also found noticeably high, but the Mann Kendall analysis failed in finding statistically significant trends in the precipitation patterns. It was not possible to identify clearly a shift of the intra-annual temporal distribution of the precipitation, neither indicating a slightly delayed onset nor an anticipated end of the rainy season.

Regarding the occurrence of floods and droughts, the survey-based findings revealed a substantial differentiation among these two events. Especially concerning droughts, 86.8% of the population reported that dry periods were more frequent during the ten years ranging between 2006 and 2015, while 23.3% experienced an extreme drought event during the last two years (2014-2015) resulting in 3.93 extreme drought events in average. Flood events were reported to be less than drought events by the local population. More than 60% of the respondents stated that flood frequency did not change during the period 2006-2015, and only 10.3% of the population experienced an extreme flood event during 2014-2015 (in average 1.69 extreme flood events per household). The judgement of the local population was confirmed by the findings of the analysis of the climatic factors regarding flood events. The analysis of the extreme precipitation in past 30 years, in fact, did not report a significant increasing trend in flood events. Similar conclusions, but this time in disagreement with the impressions of the local population, were derived from the analysis of the dry periods. This difference among perceived occurrence and observed droughts could be explained by the misperception of the local population confounding the observed increasingly frequent heat-wave events as more frequent droughts. However, the analysis of the meteorological (SPI-3) and agricultural (SPI-6) drought indicators, confirmed the occurrence of a number of dry periods that could be in line with the ones reported by the local population in the portion of the basin laying within the borders of Benin and Niger. On the other hand, the analysis of the SPI failed to find significant dry periods in the Beninese portion of the Mékrou.

The household survey analysis reported additional important findings regarding the measures adopted by the household of the Mékrou river basin to mitigate floods and droughts. Among a list of options the most significant household mitigation measures were identified among change of crop seeds, plantation of more trees, practicing more often non-agricultural economic activities, while, especially in Niger, a considerable part of the population migrates due to the losses in agriculture caused by

the decreased rainfall. Indicative are also the findings regarding the impacts of floods and droughts and their costs. For those households that experienced an extreme flood during 2014-2015 the most frequent impacts were reported to be crop production losses, damages to houses, and loss of livestock. At the same time, the loss of crop production, malnutrition, and loss of livestock were the most important impacts of extreme droughts. Additionally, to specifying the impacts of extreme floods and droughts, their total costs per household was estimated. The cost assessment is two-folded based on the sample estimations as well as on the results of the application of two linear multivariate regression econometric models. The average costs caused by flood events in the period 2014-2015 was estimated in 522 Euro per affected household basing on the average declared losses. Regarding the cost assessment of extreme droughts the average value of the sample was 390 Euro per household. This study confirmed the difficulty in the natural hazard cost estimation at the household level, even in case of recent events. A considerable percentage of the household representatives (27% for floods and 38% for droughts) were not able to provide an estimation of the costs of the extreme events that they recently experienced.

Regarding the methodological approach of this work, the combination of the household survey data analysis with the study of the climatic variables could provide an integrated assessment of floods and droughts, especially in cases like the Mékrou river, where the accessibility to reliable information is very limited. The survey approach, in particular, could provide data at household level that could be used for a detailed qualitative and quantitative assessment of natural hazards like floods and droughts. Potential limitations of this approach mainly refer to the information biases and misperceptions of the local population that could influence the objectiveness of their responses. Furthermore, such survey approach depicts the opinions at a specific time framework and, therefore, should be periodically repeated to better validate the findings. This implies the need of increased financial and human resources dedicated to this purpose. The major potential benefit of this approach is to provide information to support decision makers and local governments leading to more effective and efficient design of floods and droughts mitigation policies and measures. Most often natural hazards risk mitigation measure and policies use either climate modelling tools or socioeconomic analysis, but rarely combining both sectors. In developing countries where information is limited such a coupling approach could integrated local characteristics and perceptions into natural hazards planning policies providing more efficient mitigation measures.

**Acknowledgments**

This work is a part of the "Water for Growth and Poverty Reduction in the Mékrou" project funded by the European Commission. This project is jointly implemented by Joint Research Centre (JRC) and by the Global Water Partnership (GWP). The household survey referred to in this article was designed and implemented by the JRC and local universities from Benin, Niger and Burkina Faso. Professor Karidia Sanon from the University of the Ouagadougou (Burkina Faso) was the head of the Burkina Faso team as well as the general coordinator of the three African field teams. Euloge Agbossou and Yèkambèssoun N'Tcha M'Po from the National Water Institute (INAE) coordinated the Benin team and Professor Boureima Ousmane from the University Abdou Moumouni de Niamey was the head of the Niger team.

**Author Contributions**

Vasileios Markantonis, Celine Dondeynaz and Cesar Carmona Moreno designed the household survey and analysed the data. Fabio Farinosi, Iban Ameztoy, Marco Pastori, Luca Marletta and Abdou Ali performed the analysis of the climate data. Vasileios Markantonis, Celine Dondeynaz, Fabio Farinosi and Iban Ameztoy prepared the first draft of the manuscript. All authors discussed the results and commented on the manuscript at all stages.

**Conflicts of Interest**

The authors declare no conflict of interest.

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

**TABLES**

*Table 1. Observed climatic changes during the last 10 years: 2006-2015*

| | BENIN | | BURKINA FASO | | NIGER | | Mékrou Basin | |
|---|---|---|---|---|---|---|---|---|
| | count | % | count | % | count | % | count | % |
| Change of the rainfall quantity | | | | | | | | |
| No change | 8 | 2.4% | | | 4 | 2.2% | 12 | 1.8% |
| Less rain | 311 | 93.7% | 141 | 95.3% | 132 | 73.3% | 584 | 88.5% |
| More rain | 13 | 3.9% | 7 | 4.7% | 44 | 24.4% | 64 | 9.7% |
| Total responses | 332 | | 148 | | 180 | | 660 | |
| Distribution of the rainfall in the year | | | | | | | | |
| No change | 21 | 6.3% | | | 4 | 2.2% | 25 | 3.8% |
| Better distribution | 16 | 4.8% | 2 | 1.4% | 5 | 2.8% | 23 | 3.5% |
| Worse distribution | 295 | 88.9% | 146 | 98.6% | 171 | 95.0% | 612 | 92.7% |
| Total responses | 332 | | 148 | | 180 | | 660 | |
| More frequent droughts | | | | | | | | |
| YES | 297 | 89.5% | 147 | 99.3% | 128 | 71.5% | 572 | 86.8% |
| NO | 35 | 10.5% | 1 | 0.7% | 51 | 28.5% | 87 | 13.2% |
| Total responses | 332 | | 148 | | 179 | | 659 | |
| More frequent floods | | | | | | | | |
| YES | 50 | 15.1% | 76 | 51.4% | 130 | 72.2% | 256 | 38.8% |
| NO | 282 | 84.9% | 72 | 48.6% | 50 | 27.8% | 404 | 61.2% |
| Total responses | 332 | | 148 | | 180 | | 660 | |
| Delay in the start of the rainy season | | | | | | | | |
| YES | 248 | 74.7% | 139 | 93.9% | 161 | 89.4% | 548 | 83.0% |
| NO | 84 | 25.3% | 9 | 6.1% | 19 | 10.6% | 112 | 17.0% |
| Total responses | 332 | | 148 | | 180 | | 660 | |
| Rainy season finishes earlier | | | | | | | | |
| YES | 293 | 88.3% | 146 | 98.6% | 163 | 92.6% | 602 | 91.8% |
| NO | 39 | 11.7% | 2 | 1.4% | 13 | 7.4% | 54 | 8.2% |
| Total responses | 332 | | 148 | | 176 | | 656 | |
| Periods of extreme heat | | | | | | | | |
| No change | 34 | 10.2% | 10 | 6.8% | 15 | 8.5% | 59 | 9.0% |
| Shorter | 75 | 22.6% | 7 | 4.7% | 17 | 9.7% | 99 | 15.1% |
| Longer | 223 | 67.2% | 131 | 88.5% | 144 | 81.8% | 498 | 75.9% |
| Total responses | 332 | | 148 | | 176 | | 656 | |

*Table 2. Statistical Analysis of reported number of floods and droughts during the last 10 years*

|  | Mékrou Basin | Benin | Burkina Faso | Niger |
|---|---|---|---|---|
| *Floods* | | | | |
| *Mean* | 1.690141 | 0.727554 | 1.783784 | 3.458333 |
| *Standard Deviation* | 2.067303 | 1.158314 | 1.431143 | 2.644289 |
| *Droughts* | | | | |
| *Mean* | 3.933962 | 3.827692 | 4.797297 | 3.361963 |
| *Standard Deviation* | 2.829325 | 2.951433 | 2.722942 | 2.489043 |

*Table 3. Household mitigation measures*

| | BENIN | | BURKINA FASO | | NIGER | | Mékrou Basin | |
|---|---|---|---|---|---|---|---|---|
| | count | % | count | % | count | % | count | % |
| Change of crop seeds | | | | | | | | |
| Action taken due to temperature | | | | | | | | |
| YES | 50 | 16.2% | 29 | 19.6% | 5 | 2.8% | 84 | 13.2% |
| NO | 259 | 83.8% | 119 | 80.4% | 175 | 97.2% | 553 | 86.8% |
| Action taken due to rainfall | | | | | | | | |
| YES | 110 | 34.0% | 99 | 66.9% | 110 | 61.1% | 319 | 48.9% |
| NO | 214 | 66.0% | 49 | 33.1% | 70 | 38.9% | 333 | 51.1% |
| Terracing the soil or using other methods to protect against erosion | | | | | | | | |
| Action taken due to rainfall | | | | | | | | |
| YES | 77 | 23.8% | 26 | 17.6% | 19 | 10.6% | 122 | 18.7% |
| NO | 247 | 76.2% | 122 | 82.4% | 161 | 89.4% | 530 | 81.3% |
| Plantation of more trees | | | | | | | | |
| Action taken due to temperature | | | | | | | | |
| YES | 84 | 27.2% | 33 | 22.3% | 8 | 4.4% | 125 | 19.6% |
| NO | 225 | 72.8% | 115 | 77.7% | 172 | 95.6% | 512 | 80.4% |
| Action taken due to rainfall | | | | | | | | |
| YES | 112 | 34.6% | 39 | 26.4% | 51 | 28.3% | 202 | 31.0% |
| NO | 212 | 65.4% | 109 | 73.6% | 129 | 71.7% | 450 | 69.0% |
| Emigration of at least one household member | | | | | | | | |
| Action taken due to rainfall | | | | | | | | |
| YES | 53 | 16.4% | 12 | 8.1% | 46 | 25.6% | 111 | 17.0% |
| NO | 271 | 83.6% | 136 | 91.9% | 134 | 74.4% | 541 | 83.0% |
| Practicing more often non-agricultural activities as sources of revenue | | | | | | | | |
| Action taken due to temperature | | | | | | | | |
| YES | 100 | 32.3% | 11 | 7.4% | 32 | 17.8% | 143 | 22.4% |
| NO | 210 | 67.7% | 137 | 92.6% | 148 | 82.2% | 495 | 77.6% |
| Action taken due to rainfall | | | | | | | | |
| YES | 119 | 36.6% | 14 | 9.5% | 20 | 11.1% | 153 | 23.4% |
| NO | 206 | 63.4% | 134 | 90.5% | 160 | 88.9% | 500 | 76.6% |

*Table 4. Experienced extreme flood events during the last two years (2014-2015)*

|  | BENIN | | BURKINA FASO | | NIGER | | Mékrou Basin | |
|---|---|---|---|---|---|---|---|---|
|  | count | % | count | % | count | % | count | % |
| Floods | | | | | | | | |
| YES | 9 | 2.7% | 51 | 34.5% | 8 | 4.4% | 68 | 10.3% |
| NO | 322 | 97.3% | 97 | 65.5% | 172 | 95.6% | 591 | 89.7% |
| Droughts | | | | | | | | |
| YES | 83 | 25.5% | 66 | 44.9% | 3 | 1.7% | 152 | 23.3% |
| NO | 243 | 74.5% | 81 | 55.1% | 177 | 98.3% | 501 | 76.7% |

*Table 5. Impacts of extreme floods and droughts to the households*

|  | BENIN | BURKINA FASO | NIGER | Mékrou Basin |
|---|---|---|---|---|
|  | count | count | count | count |
| Impacts of floods | | | | |
| Damage to the house | 3 | 24 | 5 | 32 |
| Loss of agricultural production | 8 | 44 | 7 | 59 |
| Injury or death of a household member |  | 1 | 3 | 4 |
| Loss of livestock |  | 16 | 4 | 20 |
| Impacts of droughts | | | | |
| Loss of agricultural production | 74 | 66 | 3 | 143 |
| Malnutrition of at least one household person | 41 | 4 | 2 | 47 |
| Loss of livestock | 20 | 4 | 3 | 27 |

*Table 6. Estimated costs of the recent flood and droughts (in FCFA)*

|  | **Mékrou Basin** | **Benin** | **Burkina Faso** | **Niger** |
|---|---|---|---|---|
| *Costs of floods* | | | | |
| *Mean* | 324563 (~495 EURO) | 40,000 | 334,326 | 160,000 |
| *Standard Deviation* | 373159 | | 378,071 | |
| *Costs of droughts* | | | | |
| *Mean* | 256440 (~391 EURO) | 262,184 | 252,803 | 0 |
| *Standard Deviation* | 324224 | 257,676 | 362,150 | 0 |

Table 7. Selected independent variables for modelling costs of floods and droughts

| *Variable* | *Scaling* | *Correlation* |
|---|---|---|
| *ECONSTAT* | 1. Rich, 2. Relatively rich, 3. Average, 4. Below average, 5. Much worse than average | CropProdLoss: -0.0786<br>LivestockLoss: -0.134 |
| *CropProdLoss* | 0 (no impact), 1 (impact) | LivestockLoss: -0.017 |
| *LivestockLoss* | 0 (no impact), 1 (impact) | PRODCROP: 0.212 |
| *PRODCROP* | Numerical Value | |

Table 8. Costs of floods – Multivariate linear regression model

| Number of obs = 48    R-sq: 0.4828 | F-test: 7.842 | | |
|---|---|---|---|
| **Independent Variables** | Coef. | Std. Err. | P>z |
| **ECONSTAT** | -128,548.5 | 47850.09 | 0.01 |
| **CropProdLoss** | 287,269.9 | 129224.5 | 0.031 |
| **LivestockLoss** | 359,272.4 | 92753.99 | 0 |
| **cons** | 429,595.2 | 225058.3 | 0.063 |
| **Cost Estimate / household** | 256,441 (FCFA) | 390.92 (EURO) | |

Table 9. Costs of droughts - Multivariate linear regression model

| Number of obs = 98    R-sq: 0.1907 | F-test: 11.196 | | |
|---|---|---|---|
| **Independent Variables** | Coef. | Std. Err. | P>z |
| **PRODCROP** | 14.08011 | 3.643078 | 0 |
| **LivestockLoss** | 137,261.8 | 74243.4 | 0.068 |
| **_cons** | 130,367.5 | 39952.2 | 0.002 |
| **Cost Estimate / household** | 324,563 (FCFA)   494.76  (EURO) | | |

Figure 1. The Mékrou river basin and the household survey area

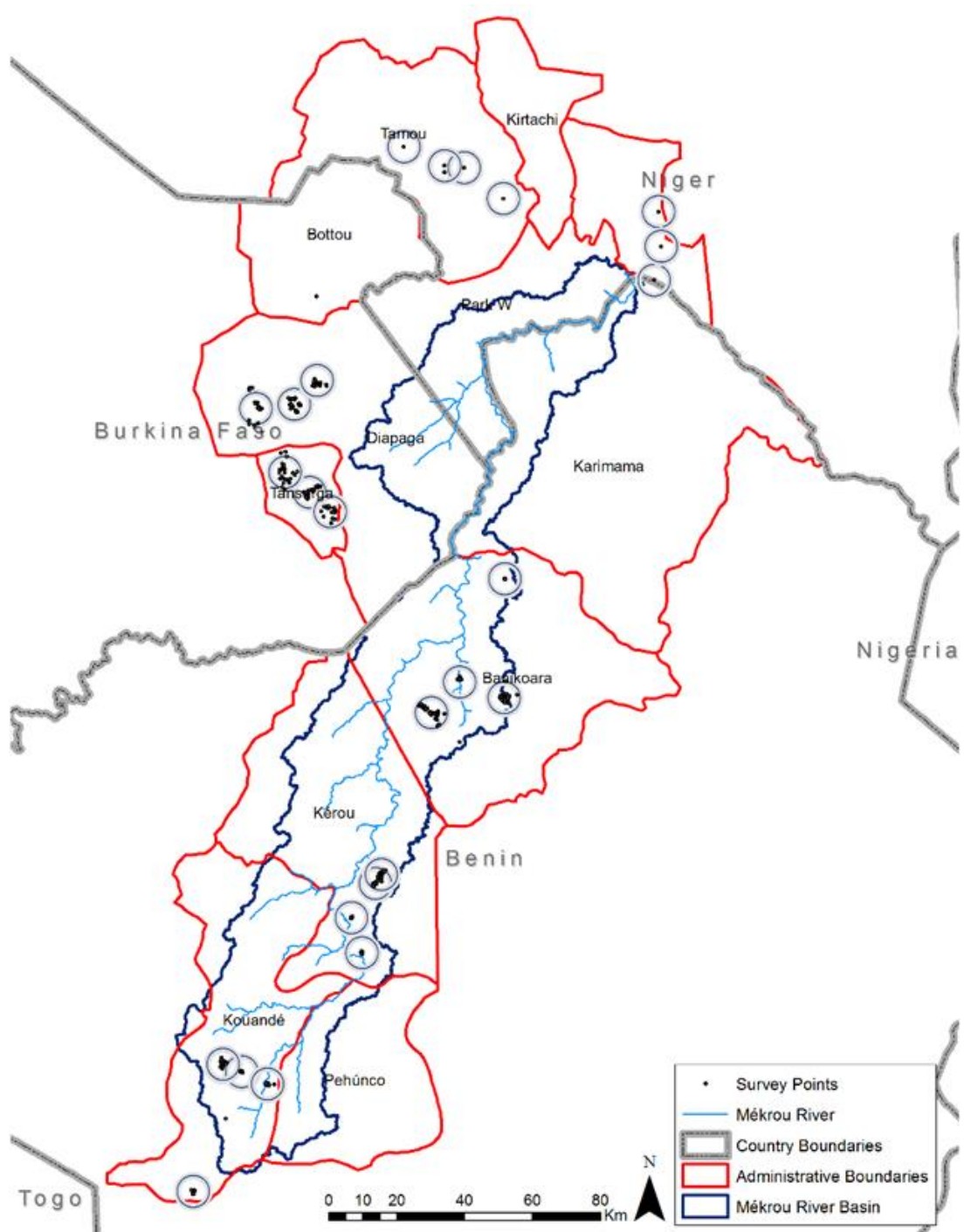

Figure 2. Comparison of seasonal distribution of precipitation for the first 25 years of the time series versus the last 10 years and Seasonal Mann-Kendall results.

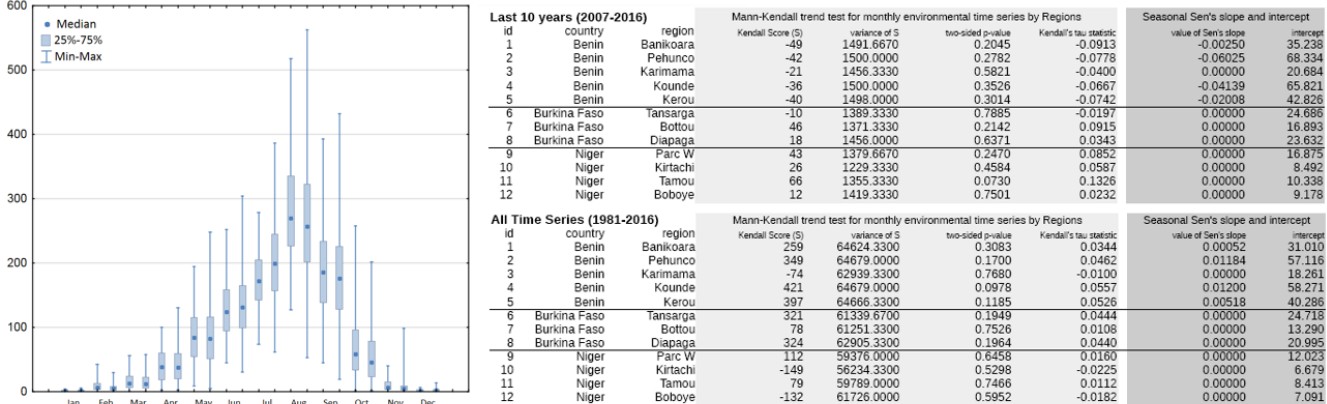

Figure 3. Temporal monthly precipitation profile for the Mékrou Area of Interest

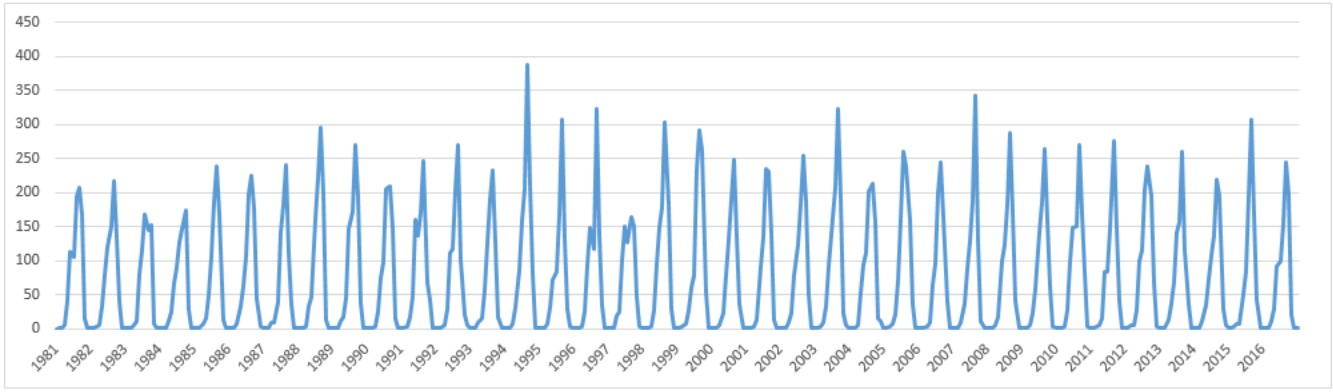

Figure 4. HWMI index computed for 1981 to 2015 on the Mékrou

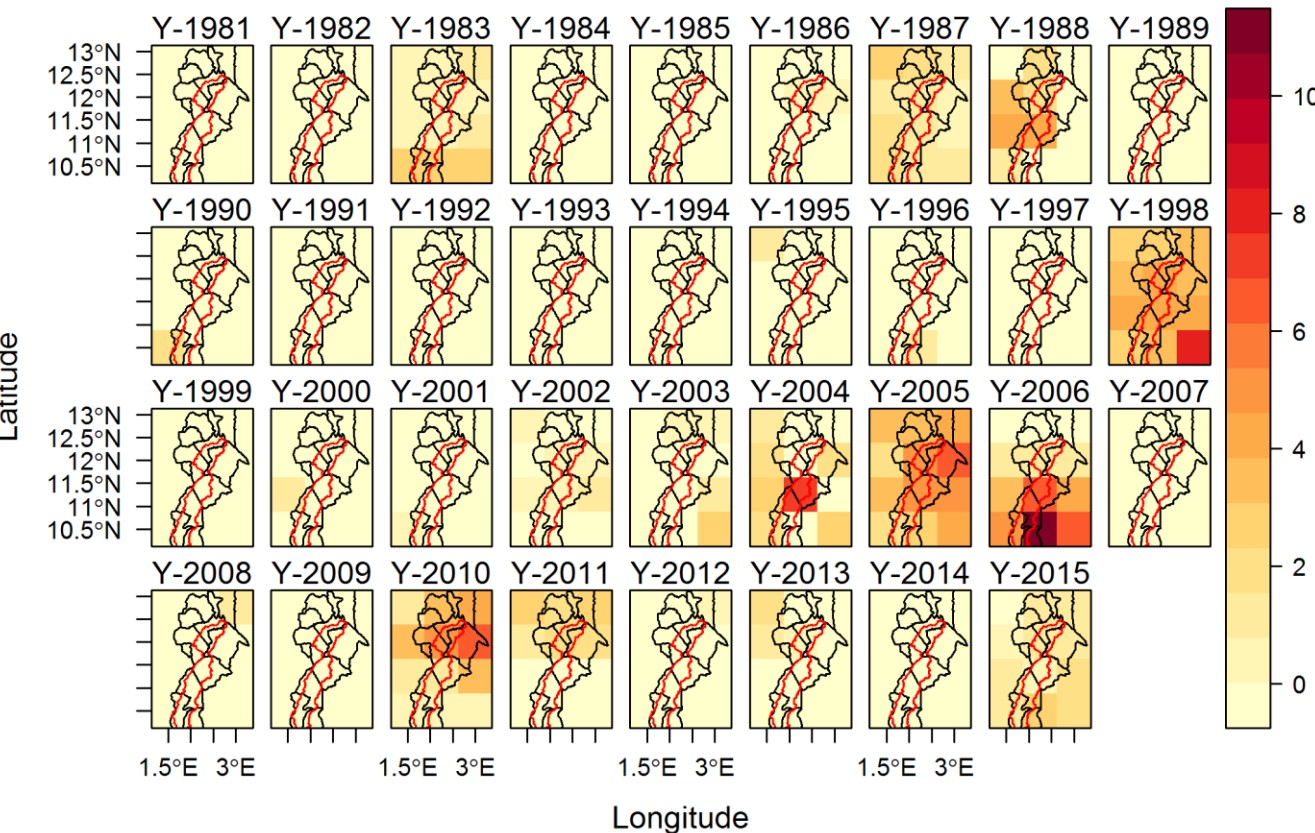

Figure 5. Mékrou river daily average discharge at Barou station as modelled in SWAT in the period 1995-2012. Dashed line indicates the average of the total period under consideration.

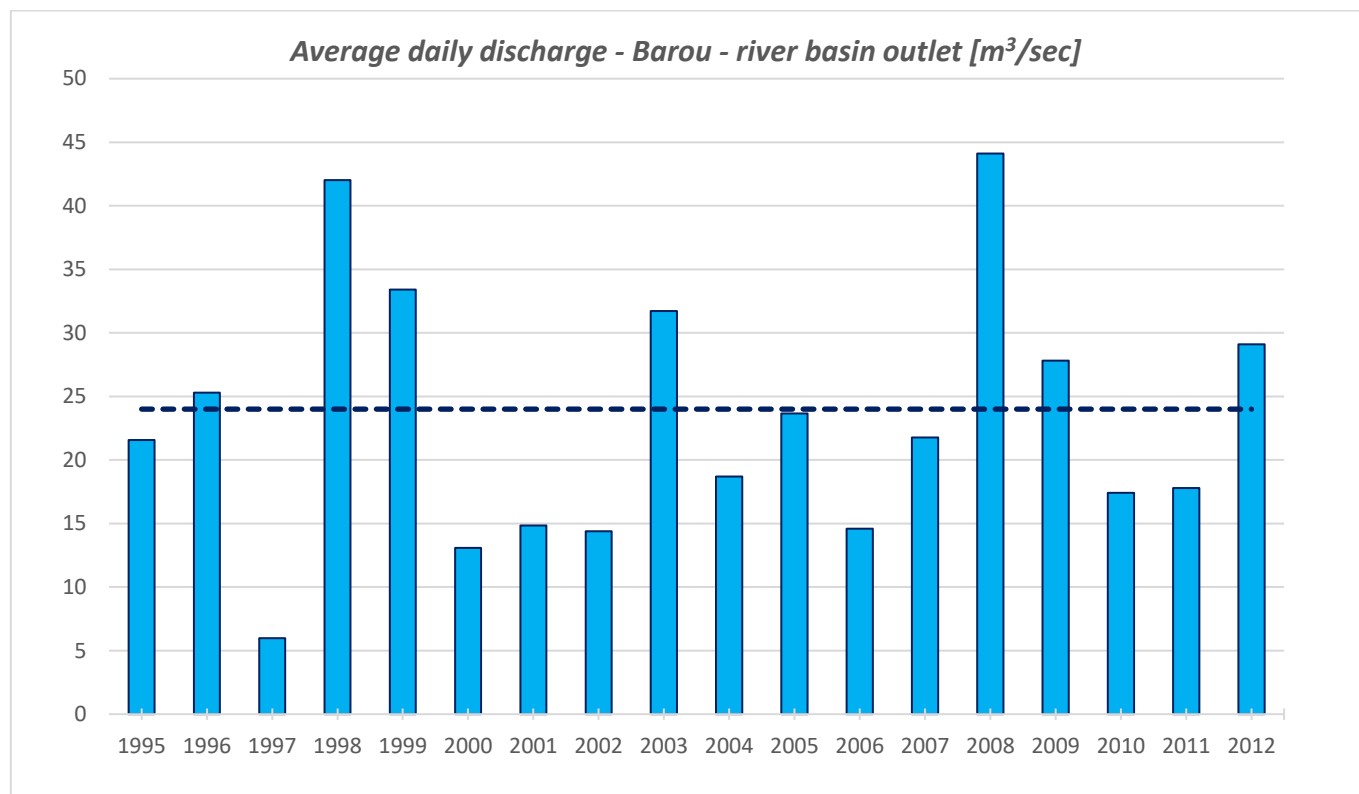

**APPENDIX A.**

QUESTIONNAIRE

