# Peer review of "Assessing floods and droughts in the Mékrou River Basin (West Africa): A combined household survey and climatic trends analysis approach"

_Natural Hazards and Earth System Sciences, 2017_

## Referee Comment (RC1) · Anonymous Referee #1 · 13 Jul 2017

**<General comments>**

My expertise allows me to evaluate only the parts of this paper that concern the socioeconomic survey. My assessments and comments about this paper, which are shown below, are solely based on how the authors perform the survey and discuss its results.

A positive aspect of this paper is the methodological novelty that it puts together original survey data in West Africa and discusses them in combination with scientific climate data. However, in the current version of the paper, methods, data and results of the survey are poorly described and presented. I also doubt that the authors have taken full advantage of the results to support their arguments. Below are the specific problems I

find in the current version of the paper.

<Specific comments>

- Descriptions of survey methods are incomplete. First, it says that the survey targeted 30 villages in three countries, but it is not clear on what criteria these villages were chosen. Do they constitute all the villages in the study region, or are they a sub-sample of the villages? If the latter is the case, how are they selected? Do they have similar geographical characteristics (elevation, vegetation, soil types, local weather conditions, etc.), or different ones? Second, how the number of surveyed households is determined for each village. Is the number proportional to the village population or not? How large is the population of each village in the first place? Third, to prove randomness of sample selection, exact methods of selecting households in each village need to be specified. Did the authors make a full list of households for each village and randomly picked up households from the list, or did they use any other methods? In the latter case, how did they warrant randomness of sampling? Fourth, was the questionnaire conducted in an in-person interview or through mail? If the former is the case, were the interviews conducted in French only or supplemented with information in a local language(s), and is there any possibility that such a linguistic choice could affect responses? Finally, response rates and summary statistics need to be presented.

- The authors would need some more analysis on the exact reasons of why perceptions of flood and drought occurrence differ across respondents. Do they reflect differences in locations of households, differences in affluence and lifestyle of households, differences in psychological biases across respondents, or simply the accuracy of responses? In particular, I suspect that detailed locational data of households have already been collected through the survey, and that it is possible to verify if differences in self-assessed occurrence of floods and droughts could be explained by differences in local weather and topological conditions or reflects other factors.

- The authors mention that obtaining cost estimates of floods and droughts from the

respondents has been difficult. In such a case, they should at least show the percentages of valid responses for the three countries, including Burkina Faso. Also, the authors would need to add some more discussions of what the cost numbers given by the respondents may really represent (costs could mean many things: asset loss, repair/resettlement costs, loss in wage and employment, loss in agricultural production, opportunity costs of labor time, medical costs, etc.) and of how accurate they are.

- Provided that estimated per-household costs of floods and droughts are credible to some extent, it may as well be useful to calculate the total costs of floods and droughts in the region, by using the information of the total number of households and of average household characteristics in the region.

СЗ

---

## Referee Comment (RC2) · Anonymous Referee #2 · 27 Aug 2017

The paper presents data for the Mekrou River Basin in West Africa from two different sources: a survey conducted in the area, covering Benin, Niger and Burkina Faso and climate data for the same region. While the idea of combining data from surveys with climate data is in principle promising, the paper has several problems:

(1) The abstract describes the datasets and the methodology, but fails to point out clearly what the subsequent study aims to show. Similarly, even after reading through the paper it is not entirely clear, what the basic message of the paper is.

(2) The literature review in the introduction is far from being exhaustive. Many important and influential studies on the impact of climate change and natural disasters on

economic development are not mentioned at all. To mention only a few:

Barrios, S., Bertinelli, L., Strobl, E., 2010, Trends in Rainfall and Economic Growth in Africa: A Neglected Cause of the African Growth Tragedy, Review of Economics and Statistics 92(2), 350-366.

Berlemann, M., Wenzel, D., 2016, Long-term Growth Effects of Natural Disasters. Empirical Evidence for Droughts, Economics Bulletin 36(1), 464-476.

Cavallo, E., Noy, I., 2011, Natural Disasters and the Economy – A Survey, International Review of Environmental and Resource Economics 5, 63-102.

Dell, M., Jones, B.F., Olken, B.A., 2014, What Do We Learn from the Weather? The New Climate– Economy Literature, Journal of Economic Literature 52, 740–798.

Felbermayr, G.J., Gröschl, J., 2014, Naturally Negative: The Growth Effects of Natural Disasters, Journal of Development Economics 111, 92-106.

Hsiang, S.M., 2010, Temperatures and cyclones strongly associated with economic production in the Caribbean and Central America, PNAS 107 (35), 15367-15372.

Skidmore, M., Toya, H., 2002, Do natural disasters promote long-run growth? Economic Inquiry 40, 664-687.

Skidmore, M., Toya, H., 2007, Economic development and the impacts of natural disasters, Economic Letters 94, 20–25.

(3) Whenever it should be a goal of the paper to contribute to the literature on evaluating the costs of climate change and natural disasters, the authors should mention other approaches existing in the literature and explain in how far the results presented in the paper are superior to these methods. As an example, the authors should refer to the Life Satisfaction Approach which has often been used to evaluate the costs of natural disasters. See e.g.

Luechinger, S., Raschky, P.A., 2009, Valuing flood disasters using the life satisfaction

Approach, Journal of Public Economics 93, 620-633.

Welsch, H., 2006, Environment and Happiness: Valuation of Air Pollution Using Life Satisfaction Data, Ecological Economics 58, 801–813.

Welsch, H., Kühling, J., 2009, Using Happiness Data for Environmental Valuation: Issues and Applications, Journal of Economic Surveys 23, 385–406.

(4) The authors claim that they combine climate data and survey data. However, I did not really understand where they are really combined. In section 3.1.4 the authors report on the population perception on the occurrence of extreme events and climate variability. However, the authors simply report the outcomes of their survey, here, without confronting the perceptions with reality (as measured by climate data). In section 3.4 the authors present some regression analysis where the reported costs of floods and droughts are related to some other variables. However, again the climate variables seem not play any role herein. It seems as the authors only discuss the two sorts of data in the same article without combining them in a meaningful way.

(5) The regression analysis in section 3.4 is conducted and/or reported very poorly. First, it remains completely unclear, why the regression analysis is conducted at all. As the result of the analysis the authors simply report the "average cost of floods per household" and "the estimated cost of droughts per household that experienced droughts". Apart from the fact that both formulations are very imprecise it is completely unclear why a regression analysis has to be conducted to find out about the costs as they are directly reported in the survey. Maybe the goal is to find out which factors determine the magnitude of the costs of affected households. But then the authors should state this clearly and discuss the hypotheses they want to test. They also make any attempt to present theoretical arguments explaining which variables should enter the regression equation. Even the variables used in the regression are explained poorly. The variable ECONSTAT seems to describe the households' wealth. However, as the variable is not metric, it makes little sense to include it in a linear regression. Whenever it shall be used, the categories should enter the regression equation as dummies. I also do not understand why the first regression includes no constant while the second one does. Again, there is no explanation. The authors do neither report a measure of the goodness of fit (such as r-square) nor the results of an F-test, as it is usual in regression analyses. The authors also seem to neglect possible heteroscedasticity, a problem occurring in almost all linear regression models. And finally, the authors' description of the choice of variables which finally enter the model (all variables reaching a P-value of less than 0.05) does not fit to Table 9, which also contains "LivestockLoss" with a P-Value of 0.068. Altogether, the empirical analysis in section 3.4 is completely flawed.

---

## Author Comment (AC1) · 3 Oct 2017

<General comments>

My expertise allows me to evaluate only the parts of this paper that concern the socioeconomic survey. My assessments and comments about this paper, which are shown below, are solely based on how the authors perform the survey and discuss its results. A positive aspect of this paper is the methodological novelty that it puts together original survey data in West Africa and discusses them in combination with scientific climate data. However, in the current version of the paper, methods, data and results of the survey are poorly described and presented. I also doubt that the authors have

taken full advantage of the results to support their arguments. Below are the specific problems I find in the current version of the paper.

'Response: We thank the Referee for the positive evaluation of the novelty of our work and for the constructive comments that will help us to improve our manuscript. '

<Specific comments>

- Descriptions of survey methods are incomplete. First, it says that the survey targeted 30 villages in three countries, but it is not clear on what criteria these villages were chosen. Do they constitute all the villages in the study region, or are they a sub-sample of the villages? If the latter is the case, how are they selected? Do they have similar geographical characteristics (elevation, vegetation, soil types, local weather conditions, etc.), or different ones? Second, how the number of surveyed households is determined for each village. Is the number proportional to the village population or not? How large is the population of each village in the first place? Third, to prove randomness of sample selection, exact methods of selecting households in each village need to be specified. Did the authors make a full list of households for each village and randomly picked up households from the list, or did they use any other methods? In the latter case, how did they warrant randomness of sampling? Fourth, was the questionnaire conducted in an in-person interview or through mail? If the former is the case, were the interviews conducted in French only or supplemented with information in a local language(s), and is there any possibility that such a linguistic choice could affect responses? Finally, response rates and summary statistics need to be presented.

'Response: We agree that the whole survey approach was not fully described in the submitted version of the paper: the revised version will be shaped to address all the issues mentioned by the Referee, providing all the details. Villages and towns were selected to include a geographically representative sample of the study area: a small river basin shared by 3 countries. The selection process was designed to keep a balance among urban and rural settlements. The number of households was selected

proportionally to the total population of each selected village or town. Households were selected randomly, in spatial terms, and not from a list since there was no list available. Interviews were conducted in person by a team of Master students supervised by a professor for each of the countries. The students received a training before starting the survey: in this session, the questionnaire was explained and discussed with them. Since the area is francophone, all the material used was written in French. The students conducting the interview, however, were local: this aspect should have allowed to overcome any eventual language issue. Response rates and all the requested statistics will be provided. '

- The authors would need some more analysis on the exact reasons of why perceptions of flood and drought occurrence differ across respondents. Do they reflect differences in locations of households, differences in affluence and lifestyle of households, differences in psychological biases across respondents, or simply the accuracy of responses? In particular, I suspect that detailed locational data of households have already been collected through the survey, and that it is possible to verify if differences in self-assessed occurrence of floods and droughts could be explained by differences in local weather and topological conditions or reflects other factors.

'Response: We thank the Referee for this useful comment. Conducting our analysis, we tested the relation between flood and drought perception and lifestyle or behavioral characteristics. This test, however, did not highlight any statistically significant relation. On the other hand, significant differences were highlighted in relation with the location of the households. We will revise the manuscript to analytically present and discuss these results.'

- The authors mention that obtaining cost estimates of floods and droughts from the respondents has been difficult. In such a case, they should at least show the percentages of valid responses for the three countries, including Burkina Faso. Also, the authors would need to add some more discussions of what the cost numbers given by the respondents may really represent (costs could mean many things: asset loss, repair/ resettlement costs, loss in wage and employment, loss in agricultural production, opportunity costs of labor time, medical costs, etc.) and of how accurate they are.

'Response: The revised manuscript will provide more information on the response rate and valid responses about the section of the survey concerning flood and drought cost estimates. However, due to the low responses and the vague information provided, we are afraid that, basing on the available data, we could not further analyze in detail the breakdown of the floods/drought impact categories. In the revise manuscript, we will further comment the existing literature and the data collected to better highlight these aspects.'

- Provided that estimated per-household costs of floods and droughts are credible to some extent, it may as well be useful to calculate the total costs of floods and droughts in the region, by using the information of the total number of households and of average household characteristics in the region.

'Response: Aggregating the cost of extreme events considering total population and number of households in the area was one of the goals we aimed to achieve with the survey and the analysis presented in this manuscript. However, the limited response rate of the section of the survey referring to the cost of extreme events at household level, in our opinion, would not be enough to provide a robust estimation of the total cost of droughts and floods in the area. In the revised version of the paper, we will analyze and discuss this aspect in more details.'

---

## Author Comment (AC2) · 3 Oct 2017

The paper presents data for the Mekrou River Basin in West Africa from two different sources: a survey conducted in the area, covering Benin, Niger and Burkina Faso and climate data for the same region. While the idea of combining data from surveys with climate data is in principle promising, the paper has several problems:

'Response: We thank the Referee for taking time to review our work and for the constructive comments. We hope that a deep revision of the manuscript, including the modifications detailed below, could be effective in addressing all the issues highlighted.' Printer-friendly version

(1) The abstract describes the datasets and the methodology, but fails to point out clearly what the subsequent study aims to show. Similarly, even after reading through the paper it is not entirely clear, what the basic message of the paper is.

'Response: The aim of this study is to provide an assessment of flood and drought by combining remotely sensed climatic data and locally collected household information. On the one hand, we aimed at producing information about a sensitive topic, as the assessment of economic and social impacts of extreme events, in an area of the world that is poorly represented in literature. On the other hand, we wanted to compare the natural hazard perception of the local population with the actual measurements of the climatic data about the extreme events referred in the survey. Additionally, the findings of this analysis are likely to benefit local decision makers toward a more effective disaster risk management in developing countries. In the revised version of the manuscript, we will revise the abstract, as well as the rest of the paper, to better highlight our aim and scope, stressing at the same time the novelty of our contribution.'

(2) The literature review in the introduction is far from being exhaustive. Many important and influential studies on the impact of climate change and natural disasters on economic development are not mentioned at all. To mention only a few: Barrios, S., Bertinelli, L., Strobl, E., 2010, Trends in Rainfall and Economic Growth in Africa: A Neglected Cause of the African Growth Tragedy, Review of Economics and Statistics 92(2), 350-366. Berlemann, M., Wenzel, D., 2016, Long-term Growth Effects of Natural Disasters. Empirical Evidence for Droughts, Economics Bulletin 36(1), 464-476. Cavallo, E., Noy, I., 2011, Natural Disasters and the Economy – A Survey, International Review of Environmental and Resource Economics 5, 63-102. Dell, M., Jones, B.F., Olken, B.A., 2014, What Do We Learn from the Weather? The New Climate– Economy Literature, Journal of Economic Literature 52, 740–798. Felbermayr, G.J., Gröschl, J., 2014, Naturally Negative: The Growth Effects of Natural Disasters, Journal of Development Economics 111, 92-106. Hsiang, S.M., 2010, Temperatures and cyclones strongly associated with economic production in the Caribbean and Central
America, PNAS 107 (35), 15367-15372. Skidmore, M., Toya, H., 2002, Do natural disasters promote long-run growth? Economic Inquiry 40, 664-687. Skidmore, M., Toya, H., 2007, Economic development and the impacts of natural disasters, Economic Letters 94, 20–25.

'Response: We agree with the Referee that some of the vast literature about natural disasters and their impacts was overlooked in the first version of this manuscript. We will revise the introduction and the discussion sections including also the suggested references. We will make sure to review literature about the impacts of natural disaster and climate change on the economic growth of the least developed countries. '

(3) Whenever it should be a goal of the paper to contribute to the literature on evaluating the costs of climate change and natural disasters, the authors should mention other approaches existing in the literature and explain in how far the results presented in the paper are superior to these methods. As an example, the authors should refer to the Life Satisfaction Approach, which has often been used to evaluate the costs of natural disasters. See e.g. Luechinger, S., Raschky, P.A., 2009, Valuing flood disasters using the life satisfaction Approach, Journal of Public Economics 93, 620-633. Welsch, H., 2006, Environment and Happiness: Valuation of Air Pollution Using Life Satisfaction Data, Ecological Economics 58, 801–813. Welsch, H., Kühling, J., 2009, Using Happiness Data for Environmental Valuation: Issues and Applications, Journal of Economic Surveys 23, 385–406.

'Response: We are aware of the many approaches presented in literature aimed at evaluating the costs of natural hazards, such as Life Satisfaction Analysis, Cost of Illness, Contingent Valuation, Input Output Analysis, Hedonic Pricing etc. However, natural hazard impacts cost assessment was only one of the goals of this analysis. Therefore, in the first version of the manuscript, we did not provide an extensive literature review about natural disaster cost assessment. However, we agree with the Referee that, since we analyzed also this aspect, we should include some literature review about this topic. That for, in the revised version of the paper, we will provide a

**NHESSD**
natural disaster cost assessment literature review summary.'

(4) The authors claim that they combine climate data and survey data. However, I did not really understand where they are really combined. In section 3.1.4 the authors report on the population perception on the occurrence of extreme events and climate variability. However, the authors simply report the outcomes of their survey, here, without confronting the perceptions with reality (as measured by climate data). In section 3.4 the authors present some regression analysis where the reported costs of floods and droughts are related to some other variables. However, again the climate variables seem not play any role herein. It seems as the authors only discuss the two sorts of data in the same article without combining them in a meaningful way.

'Response: First of all, we would like to apologize for the error in numbering the 3.1 subsections (page 7 and 8). The whole section 3.1 was dedicated to the presentation of the results of the analyses conducted over the trends detected in: precipitation (page 7 lines 9-25): Standard Precipitation Index (page 7, line 28 to page 8, line 11); heat-waves (page 8 lines 15-20); and river flow trends (page 8, line 24 to page 9, line 5). In the same section, as mentioned by the Referee, we presented the results of the survey (page 9 lines 8-32). We agree that the results were firstly presented separately, but in the discussion section (section 4) we combined the results of the survey with the climatic trends observed. In particular we stated: "This paper combines the results of a household survey and climate data analysis to assess floods and droughts as well as climate variability in the Mékrou river basin in West Africa. The opinions and perceptions of household representatives revealed a strong climate variability at a ten years period (2006-2015). It is worth mentioning that 83% of the population, during this period, noticed a delayed onset of the rainy season. In addition, 91% of the population observed also an anticipated end of the wet season. Moreover, 88.5% of the respondents reported a general reduction of the precipitation during the ten years period under consideration, and 75.9% reported an increase in magnitude and frequency of the extreme heat events. This tendency is partially confirmed by the analysis of the
climatic variables, mainly based on precipitation and temperature data. The findings of the analysis confirmed the increase of both frequency and magnitude of the heatwaves in the area of study. The climatic variability was also found noticeably high, but the Mann Kendall analysis failed in finding statistically significant trends in the precipitation patterns. It was not possible to identify clearly a shift of the intra-annual temporal distribution of the precipitation, neither indicating a slightly delayed onset nor an anticipated end of the rainy season" (page 13, lines 1-13). In the revised version of the paper, we will further describe the combination of the survey results with the outcome of the analysis of the climate data. Regarding the regression analysis in section 3.4, we have tested how climate variables influenced the costs of floods and droughts, but unfortunately they proved to be statistically non-significant. For this reason, as explained in the methodological section (Page 4, Line 30), they were not included among the variables determining the costs. In the revise version of the study, we will include alternative compositions of the statistical model including also the non-significant variables to fully describe our findings.'

(5) The regression analysis in section 3.4 is conducted and/or reported very poorly. First, it remains completely unclear, why the regression analysis is conducted at all. As the result of the analysis the authors simply report the "average cost of floods per household" and "the estimated cost of droughts per household that experienced droughts". Apart from the fact that both formulations are very imprecise it is completely unclear why a regression analysis has to be conducted to find out about the costs as they are directly reported in the survey. Maybe the goal is to find out which factors determine the magnitude of the costs of affected households. But then the authors should state this clearly and discuss the hypotheses they want to test.

'Response: We agree with the Referee that the regression analysis could be further elaborated and better explained in both scope and nature. Our main aim was to investigate the relative importance of those factors influencing floods and droughts costs, independently of the overall estimated values. This would represent a very important
analysis for the natural hazards related decision making and planning. Beside linear regression, we tested also other approaches, including logarithmic regression, but the approach presented in the paper appeared to be the best performing for the problem under consideration. In the revised version of the paper we will provide additional statistics proving these aspects.'

They also make any attempt to present theoretical arguments explaining which variables should enter the regression equation. Even the variables used in the regression are explained poorly.

'Response: Variable selection was conducted basing on their statistical significance. As mentioned in the response to the previous comment, in the revised manuscript, we will provide alternative compositions of the statistical model in order to discuss the possible use of other variables and to demonstrate their poor significance in this context. A more detailed description of the variable used (mainly presented in table 7 in the current version) will be also provided. Additionally, theoretical considerations explaining which variables should enter the regression equation, also considering existing literature, will be included.'

The variable ECONSTAT seems to describe the households' wealth. However, as the variable is not metric, it makes little sense to include it in a linear regression. When-ever it shall be used, the categories should enter the regression equation as dummies. I also do not understand why the first regression includes no constant while the second one does. Again, there is no explanation. The authors do neither report a measure of the goodness of fit (such as r-square) nor the results of an F-test, as it is usual in regression analyses. The authors also seem to neglect possible heteroscedasticity, a problem occurring in almost all linear regression models. And finally, the authors' description of the choice of variables which finally enter the model (all variables reaching a P-value of less than 0.05) does not fit to Table 9, which also contains "LivestockLoss" with a P-Value of 0.068. Altogether, the empirical analysis in section 3.4 is completely flawed.

NHESSD
'Response: Since the dependent variable is continuous, a linear regression model is appropriate to analyze the problem under consideration. The use of categorical predictors in linear models makes perfectly sense. However, we realized that our use of the variable ECONSTAT was typical of a 2-levels dummy, while the variable includes 5 categories. We will revise our statistical model to correct this and include other possible composition of predictors. In the revised manuscript, we will provide statistics including F-test and R-square, and we will discuss the problem of heteroscedasticity. We agree with the Referee that the P-value of the variable "LivestockLoss" is (a little) higher than 0.05, however we decided to include it in the model due to the importance of the animal farming in the area under consideration. Also the significance of this variable will be tested again in the light of the deep revision of the statistical analysis proposed above.'

**NHESSD**

---

## Author Response (AR1)

**Point by point response to Reviewers comments**

**1st Reviewer**

<General comments>
My expertise allows me to evaluate only the parts of this paper that concern the socioeconomic survey. My assessments and comments about this paper, which are shown below, are solely based on how the authors perform the survey and discuss its results. A positive aspect of this paper is the methodological novelty that it puts together original survey data in West Africa and discusses them in combination with scientific climate data. However, in the current version of the paper, methods, data and results of the survey are poorly described and presented. I also doubt that the authors have taken full advantage of the results to support their arguments. Below are the specific problems I find in the current version of the paper.

'Response: We thank the Referee for the positive evaluation of the novelty of our work. We used his constructive comments to improve our manuscript. '

<Specific comments>

- Descriptions of survey methods are incomplete. First, it says that the survey targeted 30 villages in three countries, but it is not clear on what criteria these villages were chosen. Do they constitute all the villages in the study region, or are they a sub-sample of the villages? If the latter is the case, how are they selected? Do they have similar geographical characteristics (elevation, vegetation, soil types, local weather conditions, etc.), or different ones? Second, how the number of surveyed households is determined for each village. Is the number proportional to the village population or not? How large is the population of each village in the first place? Third, to prove randomness of sample selection, exact methods of selecting households in each village need to be specified. Did the authors make a full list of households for each village and randomly picked up households from the list, or did they use any other methods? In the latter case, how did they warrant randomness of sampling? Fourth, was the questionnaire conducted in an in-person interview or through mail? If the former is the case, were the interviews conducted in French only or supplemented with information in a local language(s), and is there any possibility that such a linguistic choice could affect responses? Finally, response rates and summary statistics need to be presented.

'Response: In the revised version we addressed most the issues mentioned by the Referee, providing all the details. More specific we explain the criteria for selecting the villages and towns, how we the random selection was done, the organisation of the field work and the provided material.'

- The authors would need some more analysis on the exact reasons of why perceptions of flood and drought occurrence differ across respondents. Do they reflect differences in locations of households, differences in affluence and lifestyle of households, differences in psychological biases across respondents, or simply the accuracy of responses? In particular, I suspect that detailed locational data of households have already been collected through the survey, and that it is possible to verify if differences in self-assessed occurrence of floods and droughts could be explained by differences in local weather and topological conditions or reflects other factors.

'Response: We tested the relation between flood and drought perception and lifestyle or behavioural characteristics. This test, however, did not highlight any statistically significant relation. The only differences occur among the three countries and these are highlighted in the revised version.'

- The authors mention that obtaining cost estimates of floods and droughts from the respondents has been difficult. In such a case, they should at least show the percentages of valid responses for the three countries, including Burkina Faso. Also, the authors would need to add some more discussions of what the cost numbers given by the respondents may really represent (costs could mean many things: asset loss, re-pair/ resettlement costs, loss in wage and employment, loss in agricultural production, opportunity costs of labor time, medical costs, etc.) and of how accurate they are.

'Response: The revised manuscript provides more information on the response rate and valid responses about the section of the survey concerning flood and drought cost estimates. Based on the available data, we could not further analyze in detail the breakdown of the floods/drought impact categories.'

- Provided that estimated per-household costs of floods and droughts are credible to some extent, it may as well be useful to calculate the total costs of floods and droughts in the region, by using the information of the total number of households and of average household characteristics in the region.

'Response: Aggregating the cost of extreme events considering total population and number of households in the area was one of the goals we aimed to achieve with the survey and the analysis presented in this manuscript. However, the limited response rate of the section of the survey referring to the cost of extreme events at household level, in our opinion, would not be enough to provide a robust estimation of the total cost of droughts and floods in the area.'

**2nd Reviewer**

The paper presents data for the Mekrou River Basin in West Africa from two different sources: a survey conducted in the area, covering Benin, Niger and Burkina Faso and climate data for the same region. While the idea of combining data from surveys with climate data is in principle promising, the paper has several problems:

'Response: We thank the Referee for taking time to review our work and for the constructive comments. We hope that the revised version effectively addressees all the issues highlighted.'

(1) The abstract describes the datasets and the methodology, but fails to point out clearly what the subsequent study aims to show. Similarly, even after reading through the paper it is not entirely clear, what the basic message of the paper is.

'Response: The aim of this study is to provide an assessment of flood and drought by combining remotely sensed climatic data and locally collected household information. On the one hand, we aimed at producing information about a sensitive topic, as the assessment of economic and social impacts of extreme events, in an area of the world that is poorly represented in literature. On the other hand, we wanted to compare the natural hazard perception of the local population with the actual measurements of the climatic data about the extreme events referred in the survey. Additionally, the findings of this analysis are likely to benefit local decision makers toward a more effective disaster risk management in developing countries. In the revised version of the manuscript, we revised the abstract, as well as the rest of the paper, to better highlight our aim and scope, stressing at the same time the novelty of our contribution.'

(2) The literature review in the introduction is far from being exhaustive. Many important and influential studies on the impact of climate change and natural disasters on economic development are not mentioned at all. To mention only a few:

Barrios, S., Bertinelli, L., Strobl, E., 2010, Trends in Rainfall and Economic Growth in Africa: A Neglected Cause of the African Growth Tragedy, Review of Economics and Statistics 92(2), 350-366.
Berlemann, M., Wenzel, D., 2016, Long-term Growth Effects of Natural Disasters. Empirical Evidence for Droughts, Economics Bulletin 36(1), 464-476.
Cavallo, E., Noy, I., 2011, Natural Disasters and the Economy – A Survey, International Review of Environmental and Resource Economics 5, 63-102.
Dell, M., Jones, B.F., Olken, B.A., 2014, What Do We Learn from the Weather? The New Climate– Economy Literature, Journal of Economic Literature 52, 740–798.
Felbermayr, G.J., Gröschl, J., 2014, Naturally Negative: The Growth Effects of Natural Disasters, Journal of Development Economics 111, 92-106.
Hsiang, S.M., 2010, Temperatures and cyclones strongly associated with economic production in the Caribbean and Central America, PNAS 107 (35), 15367-15372.
Skidmore, M., Toya, H., 2002, Do natural disasters promote long-run growth? Economic Inquiry 40, 664-687.
Skidmore, M., Toya, H., 2007, Economic development and the impacts of natural disasters, Economic Letters 94, 20–25.

'Response: We revised the introduction and the discussion sections as well as we provided a more thorough literature review about the impacts of natural disaster and climate change on the economic growth of the least developed countries. '

(3) Whenever it should be a goal of the paper to contribute to the literature on evaluating the costs of climate change and natural disasters, the authors should mention other approaches existing in the literature and explain in how far the results presented in the paper are superior to these methods. As an example, the authors should refer to the Life Satisfaction Approach, which has often been used to evaluate the costs of natural disasters. See e.g.

Luechinger, S., Raschky, P.A., 2009, Valuing flood disasters using the life satisfaction Approach, Journal of Public Economics 93, 620-633.
Welsch, H., 2006, Environment and Happiness: Valuation of Air Pollution Using Life Satisfaction Data, Ecological Economics 58, 801–813.
Welsch, H., Kühling, J., 2009, Using Happiness Data for Environmental Valuation: Issues and Applications, Journal of Economic Surveys 23, 385–406.

'Response: We are aware of the many approaches presented in literature aimed at evaluating the costs of natural hazards, such as Life Satisfaction Analysis, Cost of Illness, Contingent Valuation, Input Output Analysis, Hedonic Pricing etc. However, natural hazard impacts cost assessment was only one of the goals of this analysis. Therefore, in our manuscript we do not provide an extensive literature review about natural disaster cost assessment.'

(4) The authors claim that they combine climate data and survey data. However, I did not really understand where they are really combined. In section 3.1.4 the authors report on the population perception on the occurrence of extreme events and climate variability. However, the authors simply report the outcomes of their survey, here, without confronting the perceptions with reality (as measured by climate data). In section 3.4 the authors present some regression analysis where the reported costs of floods and droughts are related to some other variables. However, again the climate variables seem not play any role herein. It seems as the authors only discuss the two sorts of data in the same article without combining them in a meaningful way.

'Response: First of all, we corrected the error in numbering the 3.1 subsections. We agree that the results were firstly presented separately, but in the discussion section (section 4) we combined the results of the survey with the climatic trends observed. In particular we stated: "This paper combines the results of a household survey and climate data analysis to assess floods and droughts as well as climate variability in the Mékrou river basin in West Africa.' In

the revised version of the paper, we further describe the combination of the survey results with the outcome of the analysis of the climate data. Regarding the regression analysis in section 3.4, we have tested how climate variables influenced the costs of floods and droughts, but unfortunately they proved to be statistically non-significant. For this reason, as explained in the methodological section, they were not included among the variables determining the costs.'

(5) The regression analysis in section 3.4 is conducted and/or reported very poorly. First, it remains completely unclear, why the regression analysis is conducted at all. As the result of the analysis the authors simply report the "average cost of floods per household" and "the estimated cost of droughts per household that experienced droughts". Apart from the fact that both formulations are very imprecise it is completely unclear why a regression analysis has to be conducted to find out about the costs as they are directly reported in the survey. Maybe the goal is to find out which factors determine the magnitude of the costs of affected households. But then the authors should state this clearly and discuss the hypotheses they want to test.

'Response: We agree with the Referee that the regression analysis could be further elaborated and better explained in both scope and nature. In the revised version we state clearly that our main aim was to investigate the relative importance of those factors influencing floods and droughts costs, independently of the overall estimated values. This would represent a very important analysis for the natural hazards related decision making and planning. Beside linear regression, we tested also other approaches, including logarithmic regression, but the approach presented in the paper appeared to be the best performing for the problem under consideration.'

They also make any attempt to present theoretical arguments explaining which variables should enter the regression equation. Even the variables used in the regression are explained poorly.

'Response: The selection of the variables was conducted basing on their statistical significance and each one is explained sufficiently'

The variable ECONSTAT seems to describe the households' wealth. However, as the variable is not metric, it makes little sense to include it in a linear regression. When-ever it shall be used, the categories should enter the regression equation as dummies. I also do not understand why the first regression includes no constant while the second one does. Again, there is no explanation. The authors do neither report a measure of the goodness of fit (such as r-square) nor the results of an F-test, as it is usual in regression analyses. The authors also seem to neglect possible heteroscedasticity, a problem occurring in almost all linear regression models. And finally, the authors' description of the choice of variables which finally enter the model (all variables reaching a P-value of less than 0.05) does not fit to Table 9, which also contains "LivestockLoss" with a P-Value of 0.068. Altogether, the empirical analysis in section 3.4 is completely flawed.

'Response: Since the dependent variable is continuous, a linear regression model is appropriate to analyze the problem under consideration. The use of categorical predictors in linear models makes perfectly sense. In the revised manuscript, we provided statistics including F-test and R-square. We agree with the Referee that the P-value of the variable "LivestockLoss" is (a little) higher than 0.05, however we decided to include it in the model due to the importance of the animal farming in the area under consideration.'

**List of Changes**

Page1: Abstract has been revised and adjusted to the specific objectives of the manuscript.

Page 1 and 2: The literature review has been thoroughly revised and improved according to the comments of both reviewers.

Page 4: The whole survey process has been rewritten and more information has been provided regarding the sample selection and the administration of the questionnaires.

Page 5: The field that concerns the design and methodology of statistical and econometric analysis has been more analytically described and justified.

Page 11 and 12: The analysis regarding the impacts of the floods and droughts as well as the statistical sample based estimation of floods and droughts costs have been reviewed and better explained.

Page 13: The objective of the econometric models has been better defined and more analytical information has been inserted in regards to variables, coherence of the models, R-square and F-test.

Page 15: The conclusions have been further developed.

General: The language has been checked and several mistakes were corrected

[revised manuscript text omitted]

**APPENDIX A.**

QUESTIONNAIRE

---

## Referee Report (RR1)

I don't think that the authors adequately addressed my comments on their previous version. Below are the points that I still find problematic.

- Some new information has been provided on the sampling method, but key information is still missing. How many villages and towns (rural and urban settlements?) exist in the regions, and how many people are living there? Did they conduct a stratified random sampling, or a different method? How high was the response rate, meaning the percentage of households that did not refuse to answer? When a household refused to participate in the survey, how did the interviewer find an alternative household? – this is important because it could lead to a selection bias
- Summary statistics (a standard table of the number of observations, the mean, the standard deviation, etc.) need to be given for demographic characteristics of the respondents
- I understand that there is no breakdown information on the cost estimates of floods and droughts from the survey, but some more intuitions still need to be given about what these cost figures may or may not include in the context of the studied areas. Right now, it is hard for the reader to interpret these numbers in any ways as there is hardly any hint of what they really mean.

---

## Referee Report (RR2)

Journal: NHESS
Title: **Assessing floods and droughts in the Mékrou River Basin (West Africa): A combined household survey and climatic trends analysis approach**
Author(s): V. Markantonis et al.
MS No.: nhess-2017-195
MS Type: Research Article
**Iteration: Second review**

The paper aims at assessing the occurrence of floods and droughts events in the Mékrou river basin, as well as at estimating damage costs at the household level and mitigation behaviours adopted by the population due these kinds of events. To this aim, it combines a quantitative approach for detecting hydro-meteorological hazard prone areas (through the analysis of gridded climate datasets) with a quali-quantitative analysis of a household survey.

The paper is well structured, the research question is clear, methods are appropriately described but results presentation and discussion is still poor and imprecise. In general, I would say that, although the paper has significantly improved with respect to its first version and major concerns raised by the two previous referees were mainly addresses, it still suffers of several technical imprecisions (also in the English Grammar) and minor criticisms that prevent its publication in the present form.

In the following specific comments are provided.

**Specific comments**

Introduction

In general, the literature review could be better organised to improve the comprehensibility of the manuscript.

*Pg. 2 line 1*
"A recent study (Shiferaw et al 2014), for instance, found that frequent drought conditions have limited the economic growth of many African countries and frustrated the benefits derived from development strategies implemented in other economic sectors," → Which other sectors? Not clear

*Pg. 2 line 7*
"For these reasons, in the recent past, the disaster risk management community has extensively worked on the development of methodologies aimed at monitoring the risk prone areas and the overall vulnerability of the population threatened by the hydro-meteorological hazards. Progresses have been made in the assessment of the occurrence of extremes events, their magnitude, and the expected climate change impacts. Additional efforts were directed towards the improvement of both the assessment of risk and the estimation of the direct and indirect impacts, in particular related to loss of human lives, economic activities, infrastructures, natural and man-made capital. Technical advancement efforts allowed also an improved assessment of current mitigation measures and policies" → References are required for this part

*Pg. 2 line 21*
"Regarding cost estimation and impact assessment, instead, the knowledge about losses caused by past extreme events is still limited for a detailed quantitative analysis in many of the African countries. As described in Markantonis et al. (2012 and 2013), the methodologies used in literature are various: Hedonic Pricing; Travel Cost; Cost of Illness Approach; Replacement Cost; Contingent Valuation; Choice Modeling ; and Life Satisfaction Analysis (Welsch and Kühling, 2009; Luechinger and 25 Raschky, 2009; Welsch, 2006)."

→ I guess these methodologies are not quantitative nor based on knowledge about past losses, otherwise the two sentences are discordant. Please, clarify

Section 2

Although the full questionnaire is attached to the paper, a Table summarising main information (i.e. data, parameters) collected by means of the survey can increase the comprehensibility of the manuscript and also the analyses of significant variables in Section 3.4.

*Pg. 5 line 1*
"Following the data cleaning and validation of the survey, the information collected was processed through statistical analysis including all the parameters investigated." → Which are these parameters? Please see the general comment before

*Pg. 5 line 23*
"In the frame of household survey interpretation, quantitative estimates of these events are needed to understand the underlying direct (rainfall and discharge) and indirect (heat-waves) factors that have an impact on the respondents answers." Which events are referred to? this sentence is not related to the previous and following ones.

*Pg. 6 line 16*
"Additionally, in order to identify precipitation anomalies, we calculated the Standardized Precipitation Index (hereafter SPI) proposed by McKee et.al (1993; 1995). This index could be applied over different time scales, each providing information about the impact of a given anomaly on the availability of water resources (WMO 2012)." → A better description of the index and its meaning is required for non-expert readers.

*Pg. 6 line 23*
"The results are given in units of standard deviation, indicating how far a given precipitation event is below (drier events are associated with a negative SD) or above (wetter event, positive SD) the long term normal distribution" → The sentence is not clear, please rephrase

Pg. 6 line 28
"Finally, in order to get a general overview of the anomalies in the different administrative units, we calculated the area percentage affected by each class over the entire time series" → Figure S1 should be included in the paper to increase the comprehensibility of presented results

Section 3

Please, be consistent in the units of measures and significant figures adopted in presenting the results, both within the text and in the corresponding tables

*Pg. 8 line 10*
"In the period under consideration, mean annual temperature ranged between about 30º C in the North to 26.5º C in the South. 10 April was the warmest month, with a maximum temperature of 40º C in the North and 35º C in the South. Minimum temperature in the coldest month (September) ranged between 19º C in the North and 15º C in the South, presenting the semi-arid region the highest range of variation of about 25ºC." → this part is not related to the contents of section 3.1.1. I would move this part in section 3.1.3 and change the section title into "Precipitation patterns"

*Section 3.1.4*
What the analysis of river discharge says about the occurrence of floods?

*Pg. 11 line 26*
"Only a small percentage of the respondents (68 out of in total 660 interviewed households) stated that they had not experienced any flood occurrence during the last two years (Table 4)." → The table reports the opposite, please check

*Pg. 12 line 16*
"The cost of the recent floods is higher in Burkina Faso (Table 6), where the average cost for an affected household in 334,326 FCFA (West Africa Francs) (approximately 495 Euro in 2017)." → (1) 495 euro in Table 6 refers to all the basin and not only to Burkina Faso. I guess authors want to refer to the whole basin, please check (2) are three numbers after the comma significant/required? Please check

*Pg. 13 line*
"According to the multivariate regression model, the average cost of floods per household during the flood events of the last two years (2014-2015) was equal to 390.92 euro" → (1) Data from the survey were mainly reported in FCFA. Please, be consistent to allow comparison (2) are two numbers after the comma significant/required? Please check

Section 4

*Pg. 14 line 14*
"Especially concerning droughts, 86.8% of the population reported that dry periods were more frequent during the 15 ten years ranging between 2006 and 2015, while 23.3% experienced an extreme drought event during the last two years (2014-2015) resulting in 3.93 extreme drought events in average." → In Section 3.1.5 authors report 4 droughts events in average. Please be consistent with significant figures in order to allow comparison and increase comprehensibility

*Pg. 15 line 7*
"522 Euro" → I do not understand what this datum refers to, please check

*Pg. 15 line 5*
"The cost assessment is two-folded based on the sample estimations as well as on the results of the application of two linear multivariate regression econometric models. The average costs caused by flood events in the period 2014-2015 was estimated in 522 Euro per affected household basing on the average declared losses. Regarding the cost assessment of extreme droughts, the average value of the sample was 390 Euro per household" → I think that a comparison between observations and model is required along with a discussion on usability of model results. Nonetheless, given that the main reason for the model is to explain significant variables for damage costs, a comment on significant variables should be added to section 4.

*Pg. 15 line 23*
"In developing countries where information is limited such a coupling approach could integrated local characteristics and perceptions into natural hazards planning policies providing more efficient mitigation measure" → some examples should be supplied on the use/usefulness of collected information in practice, to be more explicative

Tables

In general, captions are quite generic and could be improved in order to better reflect the contents of the tables

Table 2 → Are six numbers after the comma significant/relevant? What do they mean/tell?

Table 5→ percentages should be reported beyond total count, to be coherent with the other tables and to increase the significance/understanding of data

Table 6 → (1) Are three numbers after the comma significant/relevant? What do they mean/tell? (2) commas are missing in the first column

Figures

Figure 1→ (1) Caption is missing. (2) Numbers can be added near survey points to highlight the number of surveys carried out in each point

Figure 2 → The figure is illegible in printed versions. Data in the table are not understandable without a description. I suggest to redraw the figure, representing also table data in graphs

Figure 3→ Please indicate the long-term average in the figure. The unit of measure is missing on the y axis.

References

I did not check the correspondence between the list of references and quotations in the text, as well as availability of all references, for lack of time.

---

## Author Response (AR2)

**Point by point response to Reviewers comments**

**1st Reviewer**

I don't think that the authors adequately addressed my comments on their previous version. Below are the points that I still find problematic.

*Authors Response:*
*We are thankful for the comments of the referee and as in the first revision also in the second one we have tried to address his comments as much as possible. Analytically see below:*

- Some new information has been provided on the sampling method, but key information is still missing. How many villages and towns (rural and urban settlements?) exist in the regions, and how many people are living there? Did they conduct a stratified random sampling, or a different method? How high was the response rate, meaning the percentage of households that did not refuse to answer? When a household refused to participate in the survey, how did the interviewer find an alternative household? – this is important because it could lead to a selection bias

*Authors Response:*
*The sampling approach is better described in the revised version. Addtionally we have included Table 1 that includes all the info requested by the reviewer.*

- Summary statistics (a standard table of the number of observations, the mean, the standard deviation, etc.) need to be given for demographic characteristics of the respondents

*Authors Response:*
*Table 2 has been added that included all the main demographic characteristics of the respodents.*

- I understand that there is no breakdown information on the cost estimates of floods and droughts from the survey, but some more intuitions still need to be given about what these cost figures may or may not include in the context of the studied areas. Right now, it is hard for the reader to interpret these numbers in any ways as there is hardly any hint of what they really mean.

*Authors Response:*
*As in the first review also in this one we have to repeat that unfortunately due to the relatively low incidence of cost damages among the respondents and the high rate of respondents who could not indicate a loss value although being affected by an extreme event we prefer to indicate only the average costs of floods and droughts. These values on their own are already a quite important finding in this paper and especially in the context of Africa where such estimations are rare. We tried using econometric modelling to correlate these values to other attributes of the survey but again we did not find any significance, most probably for the above mentioned reasons. In this context, trying to interpret the costs induces a high risk to provide misleading information and therefore we prefer to provide the analysis as it is.*

**2nd Reviewer**

Journal: NHESS Title: **Assessing floods and droughts in the Mékrou River Basin (West Africa): A combined household survey and climatic trends analysis approach** Author(s): V. Markantonis et al. MS No.: nhess-2017-195 MS Type: Research Article

**Iteration: Second review**

The paper aims at assessing the occurrence of floods and droughts events in the Mékrou river basin, as well as at estimating damage costs at the household level and mitigation behaviours adopted by the population due these kinds of events. To this aim, it combines a quantitative approach for detecting hydro-meteorological hazard prone areas (through the analysis of gridded climate datasets) with a quali-quantitative analysis of a household survey.

The paper is well structured, the research question is clear, methods are appropriately described but results presentation and discussion is still poor and imprecise. In general, I would say that, although the paper has significantly improved with respect to its first version and major concerns raised by the two previous referees were mainly addresses, it still suffers of several technical imprecisions (also in the English Grammar) and minor criticisms that prevent its publication in the present form.
In the following specific comments are provided.

*Authors Response:*
*We thank the reviewer for his constructive comments. Specifically concerning the language, the reviewed version has been gone through a thorough proof reading and is substantially improved. Regarding all the other comments, we provide our response one by one as follows.*

**Specific comments**

Introduction

In general, the literature review could be better organised to improve the comprehensibility of the manuscript.

*Pg. 2 line 1*
"A recent study (Shiferaw et al 2014), for instance, found that frequent drought conditions have limited the economic growth of many African countries and frustrated the benefits derived from development strategies implemented in other economic sectors," → Which other sectors? Not clear

*Authors Response:*
*The sentence was modifies as follows:*
*"A recent study (Shiferaw et al 2014), for instance, found that frequent drought conditions have limited the economic growth of many African countries and frustrated the benefits derived from development strategies implemented in the education and technological innovation sectors, confirming also the findings of a previous study (Toya and Skidmore, 2007). "*

*Pg. 2 line 7*
"For these reasons, in the recent past, the disaster risk management community has extensively worked on the development of methodologies aimed at monitoring the risk prone areas and the overall vulnerability of the population threatened by the hydro-meteorological hazards. Progresses have been made in the assessment of the occurrence of extremes events, their magnitude, and the expected climate change impacts. Additional efforts were directed towards the improvement of both the assessment of risk and the estimation of the direct and indirect impacts, in particular related to loss of human lives, economic activities, infrastructures, natural and man-made capital. Technical advancement efforts allowed also an improved assessment of current mitigation measures and policies" → References are required for this part

*Authors Response:*

*The references for the mentioned sentences were added as requested.*
*The paragraph reads now as follows:*
*"For these reasons, in the recent past, the disaster risk management community has extensively worked on the development of methodologies aimed at monitoring the risk prone areas and the overall vulnerability of the population threatened by the hydro-meteorological hazards (UNISDR, 2017, 2013, 2011, 2008, 2004). Progresses have been made in the assessment of the occurrence of extremes events, their magnitude, and the expected climate change impacts (Alfieri et al., 2012; Hallegatte, 2012). Additional efforts were directed towards the improvement of both the assessment of risk and the estimation of the direct and indirect impacts, in particular related to loss of human lives, economic activities, infrastructures, natural and man-made capital (UNISDR, 2015). Technical advancement efforts allowed also an improved assessment of current mitigation measures and policies (Bouwer et al., 2011; Bubeck and Kreibich, 2011; Green et al., 2011; Logar and van den Bergh, 2011; Markantonis et al., 2012). "*

*Pg. 2 line 21*
"Regarding cost estimation and impact assessment, instead, the knowledge about losses caused by past extreme events is still limited for a detailed quantitative analysis in many of the African countries. As described in Markantonis et al. (2012 and 2013), the methodologies used in literature are various: Hedonic Pricing; Travel Cost; Cost of Illness Approach; Replacement Cost; Contingent Valuation; Choice Modeling ; and Life Satisfaction Analysis (Welsch and Kühling, 2009; Luechinger and 25 Raschky, 2009; Welsch, 2006)." → 	I guess these methodologies are not quantitative nor based on knowledge about past losses, otherwise the two sentences are discordant. Please, clarify

*Authors Response:*
*The whole sentence has been re-written.*

Section 2

Although the full questionnaire is attached to the paper, a Table summarising main information (i.e. data, parameters) collected by means of the survey can increase the comprehensibility of the manuscript and also the analyses of significant variables in Section 3.4.

*Pg. 5 line 1*
"Following the data cleaning and validation of the survey, the information collected was processed through statistical analysis including all the parameters investigated." → 	Which are these parameters? Please see the general comment before

*Authors Response:*
*The parameters refer to all the questions included in the questionnaire; socioeconomic, mitigation measures, cost estimates etc.*

*Pg. 5 line 23*
"In the frame of household survey interpretation, quantitative estimates of these events are needed to understand the underlying direct (rainfall and discharge) and indirect (heat-waves) factors that have an impact on the respondents answers." Which events are referred to? this sentence is not related to the previous and following ones.

*Authors Response:*
*The sentence has been reframed to: "In order to ensure an efficient interpretation of household survey's results, quantitative estimates of trends in climate related variables are*

*needed to understand the underlying direct (rainfall and discharge) and indirect (heat-waves) factors that have an impact on the respondents answers."*

*Pg. 6 line 16*
"Additionally, in order to identify precipitation anomalies, we calculated the Standardized Precipitation Index (hereafter SPI) proposed by McKee et.al (1993; 1995). This index could be applied over different time scales, each providing information about the impact of a given anomaly on the availability of water resources (WMO 2012)." → A better description of the index and its meaning is required for non-expert readers.

*Authors Response:*
*A more accessible description of the Standardized Precipitation Index, directed to a non technical audience, was added as requested.The paragraph reads now as follows:*
*"Additionally, in order to identify precipitation anomalies, we calculated the Standardized Precipitation Index (hereafter SPI) proposed by McKee et.al (1993; 1995). The SPI measures the deviation of precipitation in a specific location from its long term mean and it is a widely used indicator for drought monitoring. This index could be applied over different time scales, each providing information about the impact of a given precipitation anomaly on the availability of water resources (WMO 2012). In this study, the 3 months and 6 months SPI (hereafter SPI-3 and SPI-6) were calculated, associated respectively to meteorological and agricultural droughts. Shorter time scales SPI is considered, in fact, a good indicator of variations of soil moisture, while on longer scales (up to 24 months), it could be associated with groundwater or reservoir levels variation (WMO 2012)."*

*Pg. 6 line 23*
"The results are given in units of standard deviation, indicating how far a given precipitation event is below (drier events are associated with a negative SD) or above (wetter event, positive SD) the long term normal distribution" → The sentence is not clear, please rephrase

*Authors Response:*
*The sentence was modified as follows:*
*"The results are given in units of standard deviation (SD), indicating how far a given precipitation event is below (drier events are associated with a negative SD) or above (wetter event, positive SD) the long term normal distribution of the precipitation observations in a given location. SPI values around the zero indicate a precipitation event in line with the long term precipitation in the specific period, negative (positive) values indicate precipitation levels below (above) historical values. SPI values could then be interpreted following a classification scheme where standard deviations are categorized into different classes, each associated to different levels of wet or dry anomaly. In this case we have divided the values following the seven categories proposed by Agnew (2000), where previous thresholds defined by McKee et al. (1993) were replaced by alternative classes (Table 3)."*

Pg. 6 line 28
"Finally, in order to get a general overview of the anomalies in the different administrative units, we calculated the area percentage affected by each class over the entire time series" → Figure S1 should be included in the paper to increase the comprehensibility of presented results

*Authors Response:*
*Figure S1 was transformed in a table and included in the body of the manuscript (Table 3 in the revised version of the manuscript).*

Section 3

Please, be consistent in the units of measures and significant figures adopted in presenting the results, both within the text and in the corresponding tables

*Pg. 8 line 10*
"In the period under consideration, mean annual temperature ranged between about 30º C in the North to 26.5º C in the South. 10 April was the warmest month, with a maximum temperature of 40º C in the North and 35º C in the South. Minimum temperature in the coldest month (September) ranged between 19º C in the North and 15º C in the South, presenting the semi-arid region the highest range of variation of about 25ºC." → this part is not related to the contents of section 3.1.1. I would move this part in section 3.1.3 and change the section title into "Precipitation patterns"

*Authors Response:*
*The sentence was moved as requested and the section renamed.*

*Section 3.1.4*

What the analysis of river discharge says about the occurrence of floods?

*Authors Response:*
*Unfortunately, lack of information about historical flood events did not allow to draw conclusion about possible flood propagation related with simulated discharge.*
*This was made clear in the section 3.1.4 by adding the following sentence:*
*"The analysis of the simulated river flows highlighted the large inter-annual variability of the discharge. In about 8 of the 18 years under consideration, the average discharge was estimated to be below the long term average by more than 20%, while in 4 instances the simulated discharge was estimated to exceed the average by more than 20% (Figure 5). Although the analysis of the river discharge saw daily values with river discharge unusually high respect to the average, due to lack of data about flood propagation during historical events and detailed topography, we were not able to estimate the possible number of flood events in the domain of the study."*
*Pg. 11 line 26*
"Only a small percentage of the respondents (68 out of in total 660 interviewed households) stated that they had not experienced any flood occurrence during the last two years (Table 4)." → The table reports the opposite, please check

*Pg. 12 line 16*
"The cost of the recent floods is higher in Burkina Faso (Table 6), where the average cost for an affected household in 334,326 FCFA (West Africa Francs) (approximately 495 Euro in 2017)." → (1) 495 euro in Table 6 refers to all the basin and not only to Burkina Faso. I guess authors want to refer to the whole basin, please check (2) are three numbers after the comma significant/required? Please check

*Authors Response:*
*334326 FCA was 510 euro in 2017 (1 euro = 656 FCA). It has been corrected accordingly in the text since it refers to Burkina Faso, not in the Basin. The comma in the whole document has been used to indicate below thousand. Indeed it can be confusing and therefore it has been removed from all the text and tables.*

*Pg. 13 line*

"According to the multivariate regression model, the average cost of floods per household during the flood events of the last two years (2014-2015) was equal to 390.92 euro" →   (1) Data from the survey were mainly reported in FCFA. Please, be consistent to allow comparison (2) are two numbers after the comma significant/required? Please check

*Authors Response:*
*We include all the values in FCFA as well as in Euro when referring to the total Basin values in order to have a better comparison. The two numbers after comma are significant (0.92 euro) if someone considers that some cents are spent daily in these country from individuals to feed.*

Section 4

*Pg. 14 line 14*
"Especially concerning droughts, 86.8% of the population reported that dry periods were more frequent during the 15 ten years ranging between 2006 and 2015, while 23.3% experienced an extreme drought event during the last two years (2014-2015) resulting in 3.93 extreme drought events in average." →   In Section 3.1.5 authors report 4 droughts events in average. Please be consistent with significant figures in order to allow comparison and increase comprehensibility

*Authors Response:*
*It has been corrected in section 3.1.5 where now it indicates the average of 3.93 drought events.*

*Pg. 15 line 7*
"522 Euro" →   I do not understand what this datum refers to, please check

*Authors Response:*
*It has been corrected. The average cost of floods as indicated in Table 9 is 495 euro.*

*Pg. 15 line 5*
"The cost assessment is two-folded based on the sample estimations as well as on the results of the application of two linear multivariate regression econometric models. The average costs caused by flood events in the period 2014-2015 was estimated in 522 Euro per affected household basing on the average declared losses. Regarding the cost assessment of extreme droughts, the average value of the sample was 390 Euro per household" →   I think that a comparison between observations and model is required along with a discussion on usability of model results. Nonetheless, given that the main reason for the model is to explain significant variables for damage costs, a comment on significant variables should be added to section 4.

*Authors Response:*
*We used the statistical sample analysis to estimate the costs of floods and droughts while the econometric models were applied to define the most significant variables. We think that this is already well explained in the paper.*

*Pg. 15 line 23*
"In developing countries where information is limited such a coupling approach could integrated local characteristics and perceptions into natural hazards planning policies providing more efficient

mitigation measure" → some examples should be supplied on the use/usefulness of collected information in practice, to be more explicative

*Authors Response:*
*Some examples for the use of the collected information have been added.*

Tables

In general, captions are quite generic and could be improved in order to better reflect the contents of the tables

Table 2 → Are six numbers after the comma significant/relevant? What do they mean/tell?

*Authors Response:*
*The values have been reworked and only two numbers are provided after the comma.*

Table 5→ percentages should be reported beyond total count, to be coherent with the other tables and to increase the significance/understanding of data

*Authors Response:*
*In this case, since the numbers are not too high and not distributed in many categories we selected only the absolute numbers to demonstrate the data.*

Table 6 → (1) Are three numbers after the comma significant/relevant? What do they mean/tell? (2) commas are missing in the first column

*Authors Response:*
*See above comments. Commas indicating below thousand have been removed from the whole document.*

Figures

Figure 1→ (1) Caption is missing. (2) Numbers can be added near survey points to highlight the number of surveys carried out in each point
Figure 2 → The figure is illegible in printed versions. Data in the table are not understandable without a description. I suggest to redraw the figure, representing also table data in graphs
Figure 3→ Please indicate the long-term average in the figure. The unit of measure is missing on the y axis.

*Authors Response:*
*All three Figures have been redrawn according to the reviewer's comments.*

References

I did not check the correspondence between the list of references and quotations in the text, as well as availability of all references, for lack of time.

Authors Response:
The correspondence of references has been cross-checked.

**List of Changes**

Page1: Abstract has been further revised.

Page 1 and 2: The literature review has been revised and improved according to the comments of the second reviewer.

Page 4: The survey process has been improved and more information has been provided regarding the sample selection and the administration of the questionnaires, while two tables have been added including sampling and demographics of the respodents.

Page 15: The conclusions have been further improved.

General: A thorough language proofing has been applied to the manuscript.

For more detailed changes see above how we addressed each comment of the reviewer.

**Assessing floods and droughts in the Mékrou River Basin (West Africa): A combined household survey and climatic trends analysis approach**

Vasileios Markantonis[1], Fabio Farinosi[1], Celine Dondeynaz[1], Iban Ameztoy[1], Marco Pastori[1], Luca Marletta[1], Abdou Ali[2], Cesar Carmona Moreno[1]

[1] European Commission,  Joint Research Centre, Ispra, Italy

[2] AgrHyMet, Niamey, Niger

*Correspondence to*: Vasileios Markantonis (vmarkantonis@gmail.com)

**Abstract.** The assessment of natural hazards such as floods and droughts is a complex issue that demands integrated approaches and high quality data. Especially in African developing countries, where information is limited, the assessment of floods and droughts, though an overarching issue that influences economic and social development, is even more challenging. This paper presents an integrated approach to assessing crucial aspects of floods and droughts in the transboundary Mékrou River Basin (a portion of the Niger River Basin in West Africa). combining climatic trends analysis and the findings of a household survey. The multi-variable trend analysis estimates, at the biophysical level, the climate variability and the occurrence of floods and droughts. These results are coupled with an analysis of household survey data that reveals the behaviour and opinions of  local residents regarding the observed climate variability and occurrence of flood and drought events, household mitigation measures and the impacts of floods and droughts. Based on survey data analysis, the paper provides a per-household cost estimation of floods and droughts that occurred over a two-year period (2014-2015). Furthermore, two econometric models are set up to identify the factors that influence the costs of floods and droughts to impacted households.

**1. Introduction**

Extreme meteorological events such as droughts and floods represent an important limitation for the development of the poorest countries, impacting in particular the most vulnerable portion of the population. In these countries, agriculture remains the main economic activity, and farming practices are mainly represented by rain-fed agriculture (Rosegrant et al., 2002). Agriculture, a sector extremely vulnerable to extreme events, is the main source of income and often  the only means of self-sufficiency of the poorest portions of the rural population in the least developed countries, including sub-Saharan Africa (Gautam, 2006; Hellmuth et al., 2007). Extreme events cause loss of lives, damage to dwellings and vulnerable rural infrastructures, reduction of capital stock, and agricultural and industrial production losses. They also pose a general threat to food security and development  leading to shocks in labour productivity, energy security, and political instability (Dell et al., 2014; Hsiang, 2010). Floods and droughts,

together with  other natural disasters , were found to have a particularly damaging impact on the African continent, in particular with respect to fatalities, affected population, and economic damages (Cavallo, 2011). A recent study (Shiferaw et al, 2014), for instance, found that frequent drought conditions have limited the economic growth of many African countries and frustrated the benefits derived from development strategies implemented in the education and technological innovation sectors, confirming  the findings of a previous study (Toya and Skidmore, 2007).  Reduced precipitation was found to be responsible for about 15 to 40% of the gap between  per capita Gross Domestic Product of  African economies and that of the rest of the developing world (Barrios et al., 2010). The negative impact on economic growth of the stresses due to everincreasing dry conditions and precipitation  was also confirmed  by Berlemann and Wenzzel (2015), who found droughtprone countries to be generally characterised by  lower education levels, lower saving rates, and higher fertility levels. For these reasons, in the recent past, the disaster risk management community has extensively worked on the development of methodologies that aim to monitor the riskprone areas and the overall vulnerability of the population threatened by the hydro-meteorological hazards  (UNISDR, 2017; 2013; 2011; 2008; 2004). Progress has been made in the assessment of the occurrence of extreme events, their magnitude, and the expected climate change impacts (Alfieri et al., 2012; Hallegatte, 2012). Additional efforts were directed towards the improvement of both the assessment of risk and the estimation of the direct and indirect impacts, in particular those related to loss of human lives, economic activities, infrastructure, and natural and man-made capital (UNISDR, 2015). Technical advancement efforts  allowed for an improved assessment of current mitigation measures and policies  (Bouwer et al., 2011; Bubeck and Kreibich, 2011; Green et al., 2011; Logar and van den Bergh, 2011; Markantonis et al., 2012). However, the benefits derived from the progress made in this discipline were mainly concentrated in the most developed and technically advanced countries, where information is more readily available and mitigation strategies are more likely to be effectively implemented (UNISDR, 2015). In the case of  African countries, the assessment of the physical impacts of  hazard events followed the general  technical development. As in the most advanced countries, in fact, the  occurrence of floods and droughts was assessed by applying remotesensingbased techniques, analysing precipitation and temperature records and their spatio-temporal distribution, as for instance in  a case study in Kenya carried out by Ngigi et al. (2005).

Although several quantitative and qualitative approaches have been developed in assessing costs of natural hazards, information about losses caused by past extreme events is still  limited to facilitate a detailed  cost estimation and impact assessment analysis for many  African countries. As described in Markantonis et al. (2012 2013), the methodologies used in  literature are various: Hedonic Pricing; Travel Cost; Cost of Illness Approach; Replacement Cost; Contingent Valuation; Choice Modelling; and Life Satisfaction Analysis (Welsch and Kühling, 2009; Luechinger and Raschky, 2009; Welsch, 2006). In this

study, however, it was decided to estimate the cost of natural disasters in the case study area based on the direct testimony of the affected population. Several studies used household surveys to acquire qualitative and quantitative information from the local population in order to estimate the damages of past hydro-meteorological events (as, for instance, Fitchett et al., 2016; Ologunorisa and Adeyemo, 2005). Depending on the key economic sectors of the case study area, various examples of cost and impact assessment could be listed. Among the most representative were studies analysing the following sectors: health (Schmitt et al., 2016), agriculture and food security (Ngigi et al., 2005; Shiferaw et al., 2014; Shisanya and Mafongoya, 2016), and tourism (Fitchett et al., 2016). An analysis of the hazard components and the various impact assessments conducted in the recent past allowed for the evaluation of risk mitigation measures and adaptation strategies. For instance, Shisanya & Mafongoya (2016) analysed the effectiveness of shifts in agricultural practices; Brower et al. (2009) focused on the evaluation of the installation of risk mitigation infrastructures; Oyekale (2015) analysed the implementation of more advanced forecasting and early warning systems; Halsnæs & Trærup (2009) focused on the implementation of specific integrated policies and practices for floods and droughts. In some of these studies, cost-benefit and willingness-to-pay analyses were used to add quantitative evidence to the qualitative descriptions.

The objective of this paper is to assess the occurrence of flood and drought events, estimate of damage costs at the household level, and describe the current mitigation behaviours adopted by the population of the Mékrou River Basin, a small catchment area in West Africa. It combines a quantitative approach for detecting hydro-meteorological hazard-prone areas (through the analysis of gridded climate datasets) with a qualitative and quantitative analysis of a household survey. This approach allows for the comparison of physical analyses of extreme events with human perceptions of the flood and drought phenomena in the Mékrou River Basin. The household survey provides sufficient information to estimate the impacts of natural hazards in terms of economic cost, and to present the most widely adopted mitigation behaviours.

Mékrou is a sub-basin of the Niger River, covering an area of 10,635 km², about 3% of the total Niger Basin surface, across three countries: Benin (80% of the basin territory), Burkina Faso (10%) and Niger (10%). Mean annual precipitation ranges from a maximum of about 1,300 mm in the southern region and 500 mm in the north. The wet season occurs between June and September, with average cumulated rainfall of 700 mm. Temperatures are also highly variable in space and time. The warmest and coldest months are April and September, respectively. Mean annual temperatures vary between 26°C and 30°C, reaching a maximum of 40°C and the minimum of 19°C. The Mékrou catchment is located in a temperate transitional area characterised by a wet season that peaks in August and a long dry season that spans the period December-April (Masih et al., 2014). During the wet season, the whole Niger River Basin is subject to regular floods. Extensive research has been conducted on its complex hydrology, which is characterised by a large system of lakes and

wetlands known as the Inland Delta (Bader et al., 2016; Tarpanelli et al., 2017), while less attention has been given to the Mékrou sub-basin. Flood and drought events are extremely frequent (Froidurot & Diedhou, 2017), and the impacts of climate change on their frequency and magnitude is unclear (Gautam, 2006). The Mékrou River Basin is characterised by lack of or poor infrastructural development and very low socioeconomic conditions. Agriculture is the key economic sector,. with the arable land used for food production, cattle farming, and the production of cotton. Therefore, climate variability is the main threat to the food and economic security of the area.

This paper  is structured as follows: Section 2 presents the methodological framework  regarding the development and application of the household survey (section 2.1) and the analysis of the biophysical variables (section 2.2). The findings of this integrated approach are given in Section 3. This section analytically presents the findings regarding precipitation patterns, temperature and river discharge in the Mékrou River Basin. Moreover, it includes the main findings of the household survey concerning the observed occurrence of floods, droughts and climate variability, household mitigation measures, the impacts of floods and droughts,. and  an econometric estimation of the costs of floods and droughts. Section 4 summarises the main findings of this approach, discusses its potential and limitations,. and presents the main conclusions.

**2. Methodology**

**2.1 Household survey implementation and analysis**

A household survey to evaluate several water- related dynamics, including extreme natural hazards, was designed in 2015 and conducted in early 2016 (February to April). Specific villages and towns were selected to include a geographically representative sample of the river basin that belongs to the three countries (Benin, Burkina Faso, and Niger). The selection process was designed to keep a balance between urban and rural settlements. The number of  households proportionally represents the total population of the respective villages or towns selected, and the number of households per country represents the country's population within the basin. Since there were no available lists of households including their socioeconomic conditions we have selected them randomly based on their location in the village/town keeping a distance of five households between the interviewing one. Table 1 presents the detailed sample and population of the selected villages. The survey was carried out by experts of the Joint Research Centre of the European Commission in cooperation with local universities from Benin, Niger and Burkina Faso. Interviews were conducted in person by a team of students supervised by a professor for each country. Before starting the survey, the students received training in which the questionnaire was thoroughly explained and discussed . Since the area is francophone, all the material used was written in French. The fact that the survey was conducted by students of local universities facilitated  communication with the local population, overcoming possible language and cultural barriers. Prior to the conduction of

the survey the survey country administrators have visited each selected village and informed the local authorities. This process secured the acceptance of the survey on the ground resulting to a 100% response rate. The information included in the household survey aims to retrieve opinions and observations based on personal judgement. A large section of the survey focused on identifying and assessing the impacts and costs of flood and drought events, climate variability as well as household mitigation measures. Regarding the occurrence of floods and droughts in the study area, two time periods were selected: 10 years (2006-2015) and 2 years (2014-2015). The logic behind this selection was to identify the most recent events (2 years), for which the local households  still had a fresh memory of the impacts suffered . The second time period (10 years) offers a longer assessment of climate variability and the occurrence of floods and droughts, which could potentially still be evaluated according to personal judgement. The whole questionnaire is included in Appendix A.

*Table 1. Sample and population of the survey area*

The survey process resulted in the collection of 660 randomly surveyed questionnaires retrieved from the areas of the three countries (Benin, Burkina Faso, and Niger) that are located in the Mékrou catchment (Figure 1). Specifically, 332 questionnaires were collected in 16 villages from the municipalities (called "Communes" in French) of Banikoara, Kouande and Kerou in Benin; 148 questionnaires were collected in 6 villages from the Communes of Diagaga and Tansarga in Burkina Faso; and 180 questionnaires were collected in 8 villages from the Communes of Falmey and Tamou in Niger. The total number of surveyed households offers greater than 95% significance in the statistical findings (a minimum of 400 questionnaires are required for 95% significance rate). In Table 2 we present basic information on the socioeconomic characteristics of the surveyed population.

Figure 1. The Mékrou River Basin and the household survey area

*Table 2. Socioeconomic characteristics of the surveyed sample*

Following the data cleaning and validation of the survey, the information collected was statistically analysed including all the parameters investigated in the questionnaire. The survey responses were evaluated using descriptive statistics that aggregate data both at river- basin and at country level. The findings were analysed at the river- basin scale, illustrating the differences between the three countries.

Apart from the statistical analysis, an econometric estimation was applied to identify the specific parameters that are highly correlated to the costs of droughts and floods that were incurred in the last two years of the study (2014-2015). Hence, besides the cost estimation based on the sample mean, two econometric models were set up to investigate the

determinants of flood and drought costs following a cause-effect logic. Performing a thorough multi-variate regression cost analysis of floods and droughts as stated in the survey and other covariates (such as socioeconomic characteristics of the population, impacts, and mitigation measures) led to the construction of models that could be  used to explain the costs of extreme hydro-meteorological events. Several types of regression models were tested, both linear and logarithmic. Linear multivariate regression models were finally chosen since they fitted the selected variables with a higher statistical performance level. Moreover, in order to ensure the coherence and readability of the models, only independent variables whose P-value was less than 0.05 were selected. One independent variable with a P-value greater than 0.05 was included due to its importance in the context of the analysis. , but its value (0.068) is still below the least acceptable P-value (0.1). R-square and F-test values were estimated , and correlation  among the independent variables was tested in order to avoid bias in the model due to collinearity .

**2.2 Analysis of biophysical variables: precipitation, temperature, and river discharge**

Precipitation and temperature patterns and changes are the main drivers of local populations' perception of water availability, especially in an area where the main economic activity is based on rain-fed agricultural production. In order to ensure an efficient interpretation of household survey's results, quantitative estimates of trends in climate related variables are needed to understand the underlying direct (rainfall and discharge) and indirect (heat-waves) factors that have an impact on the respondents answers. As analysis of rainfall events above and below the long-term average distribution in conjunction with inter and intra-annual analyses are useful for depicting anomaly patterns and trends. They help better understand, for example, the main drivers of meteorological droughts. This is particularly relevant in order to compare local populations' perception of climate variables with quantitative estimates. Events above and below the long-term rainfall average distribution were studied in conjunction with intra-annual precipitation analysis . To complement this, we analysed river discharge regimes, as precipitation anomalies could be translated into hydrological droughts, thus generating a water resource imbalance, reduced groundwater levels, reservoir depletion, etc. (Liu et al., 2016). Secondly, considering the increasing number of heatwaves that occurred during the past decade in Africa (Ceccherini et al. 2017), we  studied their magnitude and spatio-temporal evolution in the Mékrou Area of Influence to explain possible misperceptions that could arise in the surveys. The objective was to compare both precipitation and temperature stress with the survey results, offering empirical evidence of their concurrence, or  explaining the underlying causes of possible contrasting results.

**2.2.1 Precipitation pattern analysis**

Descriptive statistics of annual and seasonal rainfall  were analysed jointly with the results of a Seasonal Kendall test for monotonic trend (SK) applied to precipitation data derived from the Climate Hazards Group

Infrared Precipitation with Station data v. 2.0 (hereafter CHIRPS). This database has a spatial resolution of 0.05º, corresponding to approximately 5 km (at the equator)), and covers all longitudes in the latitude range 50°S-50°N spanning a time horizon included inof 1981 to the period 1981 present daysday (Funk et. al.., 2014). The SK test used is based on Hirsch and Slack (1984) and it was applied over two different time ranges: the entire time series (about 35 years), and over the last ten years, the year period considered in the household survey. (2006-2015). The test is a modified version of the previouslyprevious SK test proposed by Hirsch et al. (1982), and attempts to reduce variablesthe serial dependence, being of variables (given that seasonal precipitation data are serially correlated.). Both are based on the non-parametric Mann-Kendall test (Mann, 1945;.; Kendall, 1975,.; Warren and Gilbert, 1987). MagnitudeThe magnitude of trends are calculated by means ofusing Sen´s Slope Estimator and Kendall tauKendall's Tau as the rank correlation coefficient.

AdditionallyIn addition, in order to identify precipitation anomalies, we calculated the Standardized Precipitation Index (hereafter SPI) proposed by McKee et.al (1993; 1995). al. (1993; 1995). The SPI measures the deviation of precipitation in a specific location from its long-term mean, and is a widely used indicator for drought monitoring. This index could be applied over different time scales, each providing information about the impact of a given precipitation anomaly on the availability of water resources (WMO, 2012). In this study, the 3 months month and 6 months month SPI (hereafter SPI-3 and SPI-6) were calculated, associated respectively towith meteorological and agricultural droughts. A shorter SPI time scale is considered to be a good indicator of variations in soil moisture, while on longer scales (up to 24 months) could be associated with groundwater or reservoir variations (WMO, 2012). For the purpose of this paper, beingas the Mékrou communities' economy is mainly based on rainfedrain-fed agriculture, we focused on the precipitation anomalies overduring the wet season, presenting the SPI-3 for the period June-August (JJA), and the SPI-6 for the period included between April andto September (AMJJAS). The results are given in units of standard deviation, (SD), indicating how far a given precipitation event is below (drier events are associated with a negative SD) or above (wetter event, positive SD) the long term normal distribution. term normal distribution of the precipitation observations in a given location. SPI values of around zero indicate a precipitation event in line with the long-term precipitation of the specific period, negative (positive) values indicate precipitation levels below (above) historical values. SPI values could then be interpreted following a classification scheme where standard deviations are categorizedcategorised into different classes, each associated towith different levels of wet or dry anomalyanomalies. In this case we have divided the values following the seven categories proposed by Agnew (2000), where previous thresholds defined by McKee et al. (1993) were replaced by alternative classes (Figure S1Table 13). Finally, in order to get a general overview of the anomalies in the different administrative units, we calculated the area percentage affected by each class over the entire time series.

*Table 13. Standardized Precipitation Index categorization and associated probability of occurrence (Agnew 2000)*

**2.2.2 Heatwave analysis**

Russo et al. (2015) defined the Heat-Wave Magnitude Index daily (hereafter HWMId) as the maximum magnitude of the heatwaves in a year, where a heatwave is defined as a period equal to or greater than three consecutive days with maximum temperature above a daily threshold calculated for a 30-year reference period. The index is based on daily maximum temperatures, taking the intensity and duration of the event into account. In our case, the source used to retrieve the maximum daily temperatures is the ERA-Interim reanalysis dataset (Berrisford et al., 2011; Dee et al., 2011) available from 1979 onwards, with an approximate spatial resolution of 80 km at the equator. The index was applied to study these events for the past 35 years across the study area.

**2.2.3 Discharge in the Mékrou River**

In order to have a quantitative estimation of water availability within the Mékrou River Basin, the hydrological model SWAT (Neitsch et al., 2011) was set up and calibrated to assess annual and monthly river discharge. The SWAT model integrates all relevant eco-hydrological processes, including water flow, surface runoff, percolation, lateral flow, groundwater flow, evapotranspiration, transmission losses, nutrient transport and turn-over, vegetation growth, land use, and water management. SWAT sub-basins were delineated using the ArcSWAT interface with a Digital Elevation Model of 90 m spatial resolution, resulting in 32 sub-basins for the whole area.

The historical discharge data recorded at the Barou gauge station (outlet of the river basin) and at the Kompoungou gauge station (a draining area that accounts for about 56% of the river basin) were used for model calibration. Lack of data availability represents a huge limitation for the hydrological analysis of the basin. In addition, these discharge observations are incomplete, as they are only available for a limited number of years (1990-2000 for Barou and 2004-2013 for Kompoungou). We used the SWAT-CUP program and manual setup to calibrate the outflow of the two monitoring stations, and obtained satisfactory efficiency statistics (Moriasi et al., 2007) at monthly scales. In Barou, the NSE (Nash–Sutcliffe model efficiency coefficient; Nash and Sutcliffe, 1970) is 0.87 and linear regression $R^2$ is 0.88, and in Kompoungou, the NSE is 0.77 and $R^2$ is 0.71. We used the SWAT-modelled discharges to consider an extended time period, required in particular to take into account climate variability, and to cover areas of the basin where observations are not available.

**3. Results**

**3.1 Biophysical variables: precipitation, temperature, and river discharge**

**3.1.1. Precipitation  patterns**

Mean Annual Precipitation in the study area varies from 500 mm in the northern administrative units (Bottou, Tamou, Kirtachi

5 and Falmey), to  a maximum of about 1,000-1,300 mm upstream in the southern portion of the basin (Kouandé, Kérou

and Banikoara (Figure S1). Generally, rainfall is highly variable in time and space, and follows a cyclical trend of wet and

dry periods. The mean monthly precipitation for the wettest month (August) in the southern and northern regions varies

between 200 mm and 300 mm, respectively, with the driest months (November-February) being close to zero.

10 ~~In the period under consideration, mean annual temperature ranged between about 30º C in the North to 26.5º C in the South. April was the warmest month, with a maximum temperature of 40º C in the North and 35º C in the South. Minimum temperature in the coldest month (September) ranged between 19º C in the North and 15º C in the South, presenting the semi-arid region the highest range of variation of about 25ºC.over~~of a short- (2007-2016)

and a long-term (1981-2006), it was possible to identify a slight decrease in rainfall during August and September,

15 while a moderate increase was detected during June and July (Figure 2). However, long- and short-term Seasonal Kendall

analyses, performed at a significance level α = 5%, were not able to identify any significant trend (Figure

2). Some spatial differences were identified analysing the last 10 years of the study: in that regard, negative slopes were

detected in the southern regions of Benin.

20 Figure 2. Box-whiskers plot comparison of monthly precipitation for the first 25 years of the time series versus the last 10 years  and Seasonal Mann-Kendall results . The Seasonal Mann-Kendal test performed did not highlight significant (p-value <0.010) increasing or decreasing trends in the analysis of the two precipitation time series.

**3.1.2 Precipitation stress analysis**

25 The results of the SPI-3 and SPI-6 analyses highlighted a period of moderate to extreme droughts during the earlier

part of the 1980s; this could  also be seen in mean monthly precipitation values, where the overall rainfall

from 1981 to 1985 is noticeably less than the long-term average (Figure 3). After this period, the positive and negative

anomalies were found to be more erratic, presenting alternate series of positive and negative events, such as some

severe wet anomalies in the southern regions in 2003 and drought events in northern regions in 1997. Regarding

30 the last ten years of the study period, only few drought anomalies affected at least 40% of the administrative

areas. The northern portion of the basin in Niger (communes of Tamou and Kirtachi) was affected by a

moderate to severe drought event during 2011, while in 2014 a similar event hit the southern portion of the basin in Benin

(Banikoara, Kérou, Pehúnco and Kouandé). On the other hand, severe to extremely wet anomalies, were  predominantly recorded during the years 2003, 2005, and 2007 in the entire Mékrou Area of Influence. Summarising the results derived from the analysis of the SPI, we found that the decade considered in the household survey was mainly characterised by precipitation in line with the long-term average. The anomalies were not particularly intense and most frequently related to wet conditions when considering the meteorological precipitation stress indicator (SPI-3 : 18 wet versus 15 dry anomalies), while a predominance of dry conditions was recorded when considering the agricultural precipitation stress (SPI-6: 17 dry versus 14 wet anomalies) (Figures S2 and S3).

Figure 3. Temporal monthly precipitation profile for the Mékrou Area of Interest

**3.1.3 Heatwaves**

The spatio-temporal evolution of the  Heat-Wave Magnitude Index (HWMI) between 1981 and 2015 is represented in Figure 4. Despite some isolated events during 1987-1988 and some extremes in 1998, the analysis highlighted a constantly increasing trend in the heatwave magnitude, starting in 2004. The spatial pattern, however, is not clear, as the whole Mékrou Area of Influence was affected. In 2005, the HWMI was found to be higher in the central and northern parts of the basin, while  the highest values in 2006 were recorded in the southern part, and in the northern part in 2010 (Figures S4 and S5).

In the period under consideration, mean annual temperature ranged between about 30º C in the North to 26.5º C in the South. April was the warmest month, with a maximum temperature of 40º C in the North and 35º C in the South. Minimum temperature in the coldest month (September) ranged between 19º C in the North and 15º C in the South, presenting the semi-arid region the highest range of variation of about 25ºC.

Figure 4. HWMI index computed for 1981 to 2015 for the Mékrou River area

**3.1.4 River water flow trends**

The annual river discharge resulting from model simulation for the period 1995-2012 is presented in Figure 5. The first 5 years of the simulation period (1990-1994) were discarded to consider an adequate model spin-up period.

Figure 5. Daily average discharge of the Mékrou River at the Barou station, as modelled using SWAT for the period 1995-2012. The dashed line indicates the average of the total period under consideration.

Modelled discharge in the period under consideration averaged around 24 m³/sec, corresponding to an average annual water flow of about 760 Mm³ (ranging from 190 to 1,400 Mm³ in 1997 and 2008, respectively). The spatial distribution of the water resources follows the topography of the basin, making the headwaters (where annual discharge remains below 100 Mm³) particularly subject to inter- and intra-annual variability (Figure S7). High annual variability of the river discharge is an important issue for the sustainable water use in the basin, especially considering that agriculture is the main economic activity and the lack of water storage infrastructures in the basin. Intra-annual discharge variability in the Mékrou River Basin follows the seasonal precipitation patterns: most abundant flows are reached after the rainy season (July-November), with the peak flow being reached in the period Aug-Sep-Oct (Figure S6).

The analysis of the simulated river flows highlighted the large inter-annual variability of the discharge. In about 8 of the 18 years under consideration, the average discharge was estimated to be below the long term average by more than 20%, while in 4 instances the simulated discharge was estimated to exceed the average by more than 20% (Figure 5). Although the analysis of the river discharge saw daily values with river discharge unusually high respect to the average, due to lack of data about flood propagation during historical events and detailed topography, we were not able to estimate the possible number of flood events in the domain of the study.

**3.1.5 Analysis of the population perception on the occurrence of extreme events and climate variability**

The household representatives gave their personal opinions on the occurrence of specific extreme events in the Mékrou River Basin during the last ten years of the study period (2006-2015) (Table 4). The selection of this 10 year framework allows for a relatively mid-term assessment of past climatic events. Regarding the occurrence of droughts, 86.8% of the households declared an increasing trend in the period under consideration. This percentage is lower in Niger, but still quite high (71.5%). In addition, the vast majority of the local population (88.5%) estimated that the levels of rainfall decreased during the last ten years of the study period. This result is particularly evident in Benin and Burkina Faso (93.7% and 95.3%, respectively). Not only did the rainfall decrease but also, according to the local population (92.7%), the seasonal distribution also changed. In fact, the majority of the respondents (83%) stated that the rainy season started with a delayed onset during the last 10 years of the study period, whereas almost all respondents (91.8%) declared an earlier end to the season. Regarding heatwaves, 76% of the interviewed population (62% in Benin) stated that periods of intense heatwaves became longer during the last 10 years of the study period. Regarding the number of events experienced, an increasing number of drought events were recorded, with a general frequency of 0 to 6 events. On average, the local population experienced in average 3.93 drought events when considering the whole Mékrou River Basin (Table 3). The number is higher in Burkina Faso (4.8 events) and slightly lower in Niger (3.4 events).

Concerning flood events, opinions are less homogeneous over the whole Mékrou area. The majority (62%) of the respondents stated that flood frequency did not increase over the last 10 years of the study period. This trend is quite heterogeneous among the three countries: 84.9% of the households in Benin agreed with the majority, the responses are discordant in Burkina Faso, whereas 72.2% of the respondents in Niger found an increasing flood frequency in the period under consideration. When asked about the number of flood events experienced, almost one third of the families stated that no flood events had occurred. The remaining replies are mostly distributed among 1 (23.3%), 2 (17.4%) and 3 (13.1%) events. As for the change in the frequency of the events, this varied across the three countries. In Benin, 60.7% of the population did not experience any flood during the last 10 years of the study period. On the other hand, large numbers of flood events were recorded primarily in Niger, followed by Burkina Faso (Table 5). In Benin, an average of only 0.7 flood events was declared during the last 10 years of the study period, 1.8 in Burkina Faso and 3.5 in Niger. The combination of these results suggests that the Nigerien portion of the Mékrou River Basin was the most prone to flood events in the period under consideration. Regarding both flood and drought events, the perceptions are heterogeneous across the three countries in the basin, with no difference based on different socioeconomic characteristics. This could be explained by the homogeneity of the socioeconomic characteristics within the basin across three countries.

*Table 24. Observed climatic changes during the last 10 years of the study period: 2006-2015*

*Table 5.*
*Statistical Analysis of reported number of floods and droughts during the last 10 years of the study period*

**3.2 Household results regarding Flood and Drought Mitigation Measures**

This section provides an empirical analysis of the household mitigation measures. The respondents chose among 13 mitigation strategies aimed at coping with changes in temperature and rainfall patterns. 8 of the original 13 strategies were considered to be negligible due to a positive response rate lower than 10%, including: practicing off-season agriculture; application of more intensive irrigation; raising less livestock in order to increase crops; raising fewer small ruminants and switching to more cattle; raising less cattle and switching to camels; raising less sheep and switching to goats; adoption of specific techniques to regenerate the necessary grass cover for the livestock; and rent or mortgage land.

The mitigation measures that recorded a positive a response rate greater than 10% are: change of crop seeds; terracing the soil or using other methods to protect against erosion; planting more trees; emigration of at least one household

member;  increasing the practice of non-agricultural activities as sources of revenue (Table 6). Changing crop seeds is  quite a common strategy for adapting to rainfall changes, and is widely applied in Burkinabe and Nigerien areas (for around 60-65% of respondents),  and to a lesser extent in Benin (34%). The practice of terracing the soil to prevent erosion is mostly applied in the Beninese part of the Mékrou River Basin. The planting of more trees is a common adaptation practice across the basin, due to both changes in precipitation (31%) and changes in temperature (20%). Emigration appears as an adaptation strategy to mitigate the economic losses resulting from the impacts of changes in rainfall patterns. Around 25% of households saw one of their members emigrating due to temperature and rainfall changes.  16.4% of Beninese respondents stated that at least one member emigrated in the last 10 years of the study period due to rainfall changes. Finally, an important number of households, primarily in Benin (around 35%) and  in Niger (around 15%), increasingly practice non-agricultural activities as sources of revenue to cope with the loss of income due to both rainfall and temperature changes.

*Table 46. Household mitigation measures*

**3. Flood and Drought impacts and their cost- assessment**

The local population of the Mékrou River Basin also indicated the occurrence of extreme events during the two -year period 2014-2015. The relatively recent occurrence of these events, some of which are still ongoing, allows  a more accurate estimation of the impacts and associated costs. Only a small percentage of the respondents (68 out of the total 660 interviewed households) stated that they had not experienced any flood occurrence during the  two years (Table 57). However, the occurrence of events differs between the three countries, with most of the events being reported in Burkina Faso (34.5% of  households affected by floods). Aligning with the frequency of reported events, the impacts of  floods are proportionally more common in Burkina Faso. The most commonly reported impacts are the loss of agricultural production, damage to  houses and loss of livestock (Table 68).

 The same analysis was applied for  recent drought events. The vast majority (76.7%) stated that they had not experienced any drought event during the last two years of the study period. However, more droughts are reported compared to floods , with 152 of the total 660 households interviewed experiencing an extreme drought event. Similarly, the occurrence of droughts is heterogeneous across the three countries. Droughts are less prominent in Niger (Table 57), where only a few droughts were reported, while a quarter of the population in Benin had experienced droughts during the last two years of the study period. In Burkina Faso, almost half of the households had experienced droughts during the  two -year period. The impacts of the recent drought were mostly recorded in Burkina Faso and Benin (Table 68) presenting a different profile. In Burkina Faso the impacts

referwere exclusively tomanifested as losses of agricultural production, while in Benin still the majority indicates most of the impact of the recent droughts was felt in the loss of agricultural production but states also, as well as malnutrition and loss of livestock as result of the recent droughts. Regarding. Both the flood and drought impacts, for both floods and droughts the collected information relates recorded relate to general categories of impacts, mainly agriculture, livestock, and housing

5  without being, as it was not feasible to collect more in--depth qualitative characteristics of the impacts.

*Table 4. Experienced extreme*,

*Table 57. Extreme flood events experienced during the last two years of the study period (2014-2015)*

10  *Table 68.*

*Table 5. Impacts of extreme floods and droughts to theon households*

The The estimated flood costs to households differ among the three countries. The cost of the recent floods is higherhighest in Burkina Faso (Table 679), where the average cost forto an affected household inis 334,326 FCFA (West AfricaAfrican Francs)

15  (approximately 495 510 Euroeuro in 2017). The estimated flood costs differ among the three countries whereas it is much higher in Burkina Faso (334,326 FCFA) than Benin and Niger (are 40,000 and 160,000 FCFA, respectively).. Furthermore, we observeobserved a difficulty fromamong the households to estimatein estimating the costs of the recent floods, especially in Benin and Niger, (the less affected countries,). where the majoritymost of the affected households were not able to provide a cost estimation. 20 out of 68 in total affected households in the basin were not able to provide a cost estimation of the floods

20  they experienced floods. The average cost of the recent droughts, 256,440 FCFA (~391 Euro), iseuro), was almost the same in Burkina Faso and in Benin (Table 6)79), and was lower than the average cost of the recent floods. Again, in this case a difficultyit is evident from thethat households found it difficult to estimate the costs of the recent droughts, especially in Benin where the majority (61 out of the 152 households) of the affected households were not able to provide a cost estimation. In this case, 61 out of the 152 households that were affected by droughts were not able to provide a cost estimation. Regarding

25  the geographical differentiation of the cost of both floods and droughts costs the, heterogeneity is observed among the three countries but not within the countries where costs are homogenous among the selected villages and towns. AdditionallyAlso, the small sample of the affected households affected by floods and droughts statistically does not allow theis insufficient for statistical aggregation of the estimated values to the total population of the basin.

30  *Table 679. Estimated costs of the recent floodfloods and droughts (in FCFA)*

**3.4 Cost assessment of floods and droughts: an econometric estimation**

AdditionallyIn addition to the statistical estimation of the cost of floods and droughts costs, two econometric models were developed to provide a reasoningan estimate of costs withbased on socioeconomic and other relative factors. A wide series of

independent variables of the household, such as socioeconomic conditions and mitigation measures , were used to find the determinants of the cost of extreme events  and to estimate the cost of floods and droughts. Using several regression models and a combination of independent variables, two models were set up that exclusively included statistically significant independent variables (P-value less than 0.05, and in one case slightly higher than 0.05 but less than 0.1). Table 810 presents the total of the independent variables, including their scaling and  correlation. The latter is important for the coherence of the multivariate models, since low correlation among the independent variables excludes the existence of multicollinearity.

*Table 810. Selected independent variables for modelling the costs of floods and droughts*

Regarding floods, the multivariate regression model in Table 911 includes  the self-stated economic status (ECONSTAT) of  households as an independent variable and as a qualitative alternative to  household income. The model reveals a strong economic status effect, meaning that the richer the household, the higher  the economic impact of floods. The negative sign is due to the structure of the variable scaling (where 1 represents rich and 5 represents significantly poorer economic conditions than the other households). Two of the main flood impacts, loss of crop productivity (CropProdLoss) and loss of livestock (LivestockLoss) were included in the model as independent variables, and were found to be significant. According to the multivariate regression model, the average cost of floods per household during the flood events of the last two years of the study period (2014-2015) was equal to 390.92 euro. In order to avoid problems of multicollinearity, correlation among the independent variables was tested. The calculated Pearson's R values, all below the 0.3 threshold, suggest a low correlation among the independent variables.

*Table 911. Costs of floods – Multivariate linear regression model*

Similarly, a multivariate regression model was applied to estimate the costs of droughts (Table 02). The independent variable related to loss of livestock was found to have a strong effect in this model too. However, drought costs are found to significantly depend on the total crop production of the households (PRODCROP). According to this regression model, the estimated cost of droughts per household that experienced drought events during the last two years of the study period (2014-2015) was 494.76 euro. Similarly low correlations were detected among the independent variables.

*Table 02. Costs of droughts - Multivariate linear regression model*

**4 Discussion and Conclusions**

This paper combines the results of a household survey and climate data analysis to assess floods . droughts and climate variability in the Mékrou River Basin in West Africa. The opinions and perceptions of household representatives revealed  strong climate variability over a ten -year period (2006-2015). It is worth mentioning that

5   83% of the population,  noticed a delayed onset of the rainy season during this period. In addition, 91% of the population also observed  an early end of the wet season. Moreover, 88.5% of the respondents reported a general reduction in precipitation during the ten -year period under consideration, and 75.9% reported an increase in  magnitude and frequency of  extreme heat events. This tendency is partially confirmed by the analysis of  climatic variables, mainly based on precipitation and temperature data. The findings of the analysis confirmed the increase in both

10   frequency and magnitude of heatwaves in the  study area. The climatic variability was also found to be noticeably high, but the Mann Kendall analysis failed to find statistically significant trends in the precipitation patterns. It was not possible to clearly identify  a shift in the intra-annual temporal distribution of  precipitation,  as there was no indication of a slightly delayed onset nor an early end of the rainy season.

15  The survey-based findings revealed a substantial difference between the occurrence of floods and droughts. With regard to droughts, 86.8% of the population reported that dry periods were more frequent during the ten years from 2006 to 2015, while 23.3% experienced an extreme drought event during the last two years (2014-2015), resulting in 3.93 extreme drought events on average. Fewer flood events  than drought events were reported by the local

20  population. More than 60% of the respondents stated that flood frequency did not change during the period 2006-2015, and only 10.3% of the population experienced an extreme flood event during the period 2014-2015 (1.69 extreme flood events on average per household). The perceptions of the local population were confirmed by the  climatic factors regarding flood events. An analysis of  extreme precipitation in the past 30 years did not report a significant increasing trend in the occurrence of flood events. Similar conclusions, but this time

25  contrary to the impressions of the local population, were derived from the analysis of  dry periods. This difference between perceived occurrence and observed droughts could be explained by the misperception of the local population confounding the observed increasingly frequent heatwave events with more frequent droughts. However, an analysis of the meteorological (SPI-3) and agricultural (SPI-6) drought indicators confirmed the occurrence of a number of dry periods that could be in line with those reported by the local population in the portion of the basin

30  located within the borders of Benin and Niger. On the other hand, the analysis of the SPI failed to find any significant dry periods in the Beninese portion of the Mékrou River Basin.

The household survey analysis reported important findings regarding the measures adopted by the households of the Mékrou River Basin to mitigate floods and droughts. From a list of options, the most significant household mitigation measures were identified as being: changing crop seeds, planting more trees, and increased practice of non-agricultural economic activities, while, especially in Niger, a considerable part of the population emigrated due to the loss in agricultural production caused by the reduced rainfall. The findings regarding the impacts of floods and droughts and their costs are also interesting. For those households that experienced an extreme flood event during the period 2014-2015, the most frequent impacts were reported to be crop production losses, damages to houses, and loss of livestock. The loss of crop production, malnutrition, and the loss of livestock were the most important impacts of extreme droughts. In addition, the total cost per household of the impacts of extreme floods and droughts was estimated. The cost assessment is two-folded based on the sample estimations and the application of two linear multivariate regression econometric models. The average cost of flood events in the period 2014-2015 was estimated at 495 euro per affected household, based on the average declared losses. The average cost of extreme droughts was 391 euro per household. This study confirmed the difficulty of estimating the costs of natural hazards at the household level, even in the case of recent events. A considerable percentage of the household representatives (27% for floods and 38% for droughts) were not able to provide an estimation of the costs of the extreme events that they recently experienced.

Regarding the methodological approach of this work, the combination of the household survey data analysis with the study of the climatic variables could provide an integrated assessment of floods and droughts, especially in cases such the Mékrou River Basin, where access to reliable information is very limited. The survey approach, in particular, could provide data at household level that could be used for a detailed qualitative and quantitative assessment of natural hazards such as floods and droughts. Potential limitations of this approach are mainly the information biases and misperceptions of the local population, which could influence the objectivity of their responses. Furthermore, such a survey approach depicts the opinions within a specific timeframe and, therefore, should be periodically repeated to better validate the findings. This would require increased financial and human resources. The major potential benefit of this approach is that it provides information to support decision-makers and local governments, leading to the more effective and efficient design of flood and drought mitigation policies and measures. Most often, natural hazards risk mitigation measures and policies use either climate modelling tools or socioeconomic analyses, but rarely combine both. In developing countries where information is limited, such a coupling approach could integrate local characteristics and perceptions into natural hazard planning policies, in order to provide more efficient mitigation measures. More specific, such an

approach could be used by state and local authorities to design risk mitigation and prevention measures and design climate adaptation strategies combining climate and socioeconomic analysis.

**Acknowledgments**

This work is a part of the "Water for Growth and Poverty Reduction in the Mékrou" project funded by the European Commission. This project is jointly implemented by the Joint Research Centre (JRC) and by the Global Water Partnership (GWP). The household survey referred to in this article was designed and implemented by the JRC and local universities from Benin, Niger and Burkina Faso. Professor Karidia Sanon from the University of  Ouagadougou (Burkina Faso) was the head of the Burkina Faso team as well as the general coordinator of the three African field teams. Euloge Agbossou and Yèkambèssoun N'Tcha M'Po from the National Water Institute (Institut National de l'Eau - INE) coordinated the Benin team, and Professor Boureima Ousmane from the  Abdou Moumouni University of Niamey was the head of the Niger team. The authors would like to thank Ms. Grainne Mulhern (JRC) for proofreading the manuscript.

**Author Contributions**

Vasileios Markantonis, Celine Dondeynaz and Cesar Carmona Moreno designed the household survey and analysed the data. Fabio Farinosi, Iban Ameztoy, Marco Pastori, Luca Marletta and Abdou Ali analysed the climate data. Vasileios Markantonis, Celine Dondeynaz, Fabio Farinosi and Iban Ameztoy prepared the first draft of the manuscript. All authors discussed the results and commented on the manuscript at all stages.

**Conflicts of Interest**

The authors declare no conflict of interest.

**TABLES**

*Table 1. Sample and population of the survey area*

| Population of the Mékrou Area of Interest | | | |
|---|---|---|---|
| Benin | Burkina Faso | Niger | Total area |
| 294921 | 79632 | 173115 | 547668 |
| **Surveyed Sample (Households)** | | | |
| Benin | Burkina Faso | Niger | Total area |
| 332 | 148 | 180 | 660 |

| Sample (number of households) by selected Communes | | | | | | |
|---|---|---|---|---|---|---|
| Benin | | | Burkina Faso | | Niger | |
| Banikorara | Kérou | Kouandé | Diapaga | Tansagra | Tamou | Birni Ngaoure |
| 160 | 80 | 92 | 95 | 53 | 100 | 80 |

| Population and Sample (number of households) by Selected Village / town | | | | |
|---|---|---|---|---|
| **Banikoara** | | | | |
| | Sampéto | Gbéniki (Kérémou) | Wangouwirou | Banikoara (town) | Total |
| Population | 1 522 | 786 | 3799 | 28402 | 32987 |
| Sample | 29 | 20 | 52 | 52 | 153 |
| **Kouande** | | | | |
| | Béket Bouramè | Mekrougourou | Goufanrou | Kouandé (town) | Total |
| Population | 1876 | 2635 | 1835 | 20723 | 27069 |
| Sample | 20 | 27 | 20 | 25 | 92 |
| **Kérou** | | | | |
| | Koussou Ouinra | Yakrigourou | Bipotoké | Kérou (town) | Total |
| Population | 2842 | 2766 | 2871 | 34246 | 42725 |
| Sample | 16 | 19 | 16 | 36 | 87 |
| **Diapaga** | | | | |
| | Mangou | Tyaga | Diapaga (town) | Total | |

| Population | 1600 | 1136 | 16000 | 18736 |
|---|---|---|---|---|
| Sample | 28 | 20 | 40 | 88 |

**Tansagra**

| | Kotchari | Lada | Tansarga (town) | Total |
|---|---|---|---|---|
| Population | 1024 | 720 | 14000 | 15744 |
| Sample | 20 | 16 | 24 | 60 |

**Birni Ngaoure**

| | Boumba | Fono Birgui | Kotaki | Flamey Djema (town) | Total |
|---|---|---|---|---|---|
| Population | 1414 | 560 | 2447 | 4467 | 8888 |
| Sample | 12 | 8 | 20 | 40 | 80 |

**Tamou**

| | Tankoune | Diney Bangou | Foulan Walagorou | Tamou (town) | Total |
|---|---|---|---|---|---|
| Population | 827 | 724 | 261 | 1827 | 3639 |
| Sample | 28 | 28 | 4 | 40 | 100 |

*Table 2. Socioeconomic characteristics of the surveyed sample*

| *Age* (sample respondents) | | | | | | | |
|---|---|---|---|---|---|---|---|
| *Benin* | | *Burkina Faso* | | *Niger* | | *Total Area* | |
| *Mean* | *StdDev* | *Mean* | *StdDev* | *Mean* | *StdDev* | *Mean* | *StdDev* |
| *41.2* | *14.9* | *44.2* | *16.2* | *49.5* | *15.3* | *44.2* | *15.7* |

| *Age Distribution* (Total Area Population based on survey aggregation) | | | | |
|---|---|---|---|---|
| | *Benin* | *Burkina Faso* | *Niger* | *Total Area* |
| *0-5* | *17.10%* | *23.50%* | *20.20%* | *19.50%* |
| *5-18* | *32.90%* | *34.40%* | *31.30%* | *32.70%* |
| *18+* | *49.90%* | *42.10%* | *48.60%* | *47.90%* |

| *Gender* (sample respondents) | | | | | | | |
|---|---|---|---|---|---|---|---|
| | *Benin* | | *Burkina Faso* | | *Niger* | | *Total Area* | |
| | *Count* | *%* | *Count* | *%* | *Count* | *%* | *Count* | *%* |
| *Male* | *226* | *68.1%* | *95* | *64.2%* | *129* | *71.7%* | *450* | *68.2%* |
| *Fenale* | *106* | *31.9%* | *53* | *35.8%* | *51* | *28.3%* | *210* | *31.8%* |

| *Education* (sample respondents) | | | | | |
|---|---|---|---|---|---|
| | *No schooling* | *No formal schooling* | *Primary school* | *Secondary school* | *Professional education* | *University* |
| *Count* | *389* | *87* | *81* | *78* | *8* | *17* |
| *%* | *58.90%* | *13.20%* | *12.30%* | *11.80%* | *1.20%* | *2.60%* |

| *Profession* (sample respondents) | | | | | |
|---|---|---|---|---|---|
| | *Unemployed* | *Self epmployed* | *public employee* | *Farmer* | *Livestock farmer* | *Other* |
| *%* | *7.70%* | *18.90%* | *2.30%* | *50.10%* | *15.90%* | *5.10%* |

| *Household Income [FCFA per month] 1 euro = 656 FCFA* (sample respondents) | | | | | |
|---|---|---|---|---|---|
| | *0 - 25000* | *25001 - 50000* | *50001 - 75000* | *75001 - 100000* | *more than 100001* | *I don't know* |
| *Count* | *200* | *100* | *77* | *37* | *414* | *121* |
| *%* | *30.50%* | *15.20%* | *11.70%* | *5.60%* | *18.60%* | *18.40%* |

*Table 13. Standardized Precipitation Index categorization and associated probability of occurrence (Agnew 2000)*

| SPI-n | Probability of Occurrence | Class |
|---|---|---|
| > 1.65 | 0.05 | Extremely Wet |
| 1.28 / 1.64 | 0.1 | Severely Wet |
| 0.84 / 1.27 | 0.2 | Moderately Wet |
| -0.84 / 0.84 | 0.5 | Normal |
| -1.28 / -0.83 | 0.2 | Moderate Drought |
| -1.65 / -1.27 | 0.1 | Severe Drought |
| < -1.65 | 0.05 | Extreme Drought |

*Table 24. Observed climatic changes during the last 10 years of the study period: 2006-2015*

| | BENIN | | BURKINA FASO | | NIGER | | Mékrou Basin | |
|---|---|---|---|---|---|---|---|---|
| | count | % | count | % | count | % | count | % |
| Change of the rainfall quantity | | | | | | | | |
| No change | 8 | 2.4% | | | 4 | 2.2% | 12 | 1.8% |
| Less rain | 311 | 93.7% | 141 | 95.3% | 132 | 73.3% | 584 | 88.5% |
| More rain | 13 | 3.9% | 7 | 4.7% | 44 | 24.4% | 64 | 9.7% |
| Total responses | 332 | | 148 | | 180 | | 660 | |
| Distribution of the rainfall in the year | | | | | | | | |
| No change | 21 | 6.3% | | | 4 | 2.2% | 25 | 3.8% |
| Better distribution | 16 | 4.8% | 2 | 1.4% | 5 | 2.8% | 23 | 3.5% |
| Worse distribution | 295 | 88.9% | 146 | 98.6% | 171 | 95.0% | 612 | 92.7% |
| Total responses | 332 | | 148 | | 180 | | 660 | |
| More frequent droughts | | | | | | | | |
| YES | 297 | 89.5% | 147 | 99.3% | 128 | 71.5% | 572 | 86.8% |
| NO | 35 | 10.5% | 1 | 0.7% | 51 | 28.5% | 87 | 13.2% |
| Total responses | 332 | | 148 | | 179 | | 659 | |
| More frequent floods | | | | | | | | |
| YES | 50 | 15.1% | 76 | 51.4% | 130 | 72.2% | 256 | 38.8% |
| NO | 282 | 84.9% | 72 | 48.6% | 50 | 27.8% | 404 | 61.2% |
| Total responses | 332 | | 148 | | 180 | | 660 | |
| Delay in the start of the rainy season | | | | | | | | |
| YES | 248 | 74.7% | 139 | 93.9% | 161 | 89.4% | 548 | 83.0% |
| NO | 84 | 25.3% | 9 | 6.1% | 19 | 10.6% | 112 | 17.0% |
| Total responses | 332 | | 148 | | 180 | | 660 | |
| Rainy season finishes earlier | | | | | | | | |
| YES | 293 | 88.3% | 146 | 98.6% | 163 | 92.6% | 602 | 91.8% |
| NO | 39 | 11.7% | 2 | 1.4% | 13 | 7.4% | 54 | 8.2% |
| Total responses | 332 | | 148 | | 176 | | 656 | |
| Periods of extreme heat | | | | | | | | |
| No change | 34 | 10.2% | 10 | 6.8% | 15 | 8.5% | 59 | 9.0% |
| Shorter | 75 | 22.6% | 7 | 4.7% | 17 | 9.7% | 99 | 15.1% |
| Longer | 223 | 67.2% | 131 | 88.5% | 144 | 81.8% | 498 | 75.9% |
| Total responses | 332 | | 148 | | 176 | | 656 | |

Table 235. *Statistical Analysis of reported number of floods and droughts during the last 10 years*

| | Mékrou Basin | Benin | Burkina Faso | Niger |
|---|---|---|---|---|
| *Floods* | | | | |
| Mean | 1.690141 | 0.7327554 | 1.783784 | 3.45833346 |
| Standard Deviation | 2.06730307 | 1.15831416 | 1.431143 | 2.644289 |
| *Droughts* | | | | |
| Mean | 3.933962 | 3.82769283 | 4.79729780 | 3.361963 |
| Standard Deviation | 2.82932583 | 2.951433 | 2.722942 | 2.48904349 |

*Table 3̶46. Household mitigation measures*

| | BENIN | | BURKINA FASO | | NIGER | | Mékrou Basin | |
|---|---|---|---|---|---|---|---|---|
| | count | % | count | % | count | % | count | % |
| Change of crop seeds | | | | | | | | |
| Action taken due to temperature | | | | | | | | 5 |
| YES | 50 | 16.2% | 29 | 19.6% | 5 | 2.8% | 84 | 13.2% |
| NO | 259 | 83.8% | 119 | 80.4% | 175 | 97.2% | 553 | 86.8% |
| Action taken due to rainfall | | | | | | | | |
| YES | 110 | 34.0% | 99 | 66.9% | 110 | 61.1% | 319 | 48.9% |
| NO | 214 | 66.0% | 49 | 33.1% | 70 | 38.9% | 333 | 51.1% |
| Terracing the soil or using other methods to protect against erosion | | | | | | | | |
| Action taken due to rainfall | | | | | | | | |
| YES | 77 | 23.8% | 26 | 17.6% | 19 | 10.6% | 122 | 18.7% |
| NO | 247 | 76.2% | 122 | 82.4% | 161 | 89.4% | 530 | 81.3% |
| Plantation of more trees | | | | | | | | |
| Action taken due to temperature | | | | | | | | |
| YES | 84 | 27.2% | 33 | 22.3% | 8 | 4.4% | 125 | 19.6% |
| NO | 225 | 72.8% | 115 | 77.7% | 172 | 95.6% | 512 | 80.4% |
| Action taken due to rainfall | | | | | | | | |
| YES | 112 | 34.6% | 39 | 26.4% | 51 | 28.3% | 202 | 31.0% |
| NO | 212 | 65.4% | 109 | 73.6% | 129 | 71.7% | 450 | 69.0% |
| Emigration of at least one household member | | | | | | | | |
| Action taken due to rainfall | | | | | | | | |
| YES | 53 | 16.4% | 12 | 8.1% | 46 | 25.6% | 111 | 17.0% |
| NO | 271 | 83.6% | 136 | 91.9% | 134 | 74.4% | 541 | 83.0% |
| Practicing more often non-agricultural activities as sources of revenue | | | | | | | | |
| Action taken due to temperature | | | | | | | | |
| YES | 100 | 32.3% | 11 | 7.4% | 32 | 17.8% | 143 | 22.4% |
| NO | 210 | 67.7% | 137 | 92.6% | 148 | 82.2% | 495 | 77.6% |
| Action taken due to rainfall | | | | | | | | |
| YES | 119 | 36.6% | 14 | 9.5% | 20 | 11.1% | 153 | 23.4% |
| NO | 206 | 63.4% | 134 | 90.5% | 160 | 88.9% | 500 | 76.6% |

Formatted
Formatted Table
Formatted
Formatted
Formatted
Formatted
Formatted
Formatted
Formatted
Formatted
Formatted
Formatted
Formatted
Formatted
Formatted
Formatted
Formatted
Formatted
Formatted
Formatted
Formatted
Formatted
Formatted
Formatted
Formatted
Formatted
Formatted
Formatted
Formatted
Formatted
Formatted
Formatted

*Table 457. Experienced extreme flood events during the last two years (2014-2015)*

|  | BENIN |  | BURKINA FASO |  | NIGER |  | Mékrou Basin |  |
|---|---|---|---|---|---|---|---|---|
|  | count | % | count | % | count | % | count | % |
| Floods |  |  |  |  |  |  |  |  |
| YES | 9 | 2.7% | 51 | 34.5% | 8 | 4.4% | 68 | 10.3% |
| NO | 322 | 97.3% | 97 | 65.5% | 172 | 95.6% | 591 | 89.7% |
| Droughts |  |  |  |  |  |  |  |  |
| YES | 83 | 25.5% | 66 | 44.9% | 3 | 1.7% | 152 | 23.3% |
| NO | 243 | 74.5% | 81 | 55.1% | 177 | 98.3% | 501 | 76.7% |

*Table 568. Impacts of extreme floods and droughts to the households*

|  | BENIN | BURKINA FASO | NIGER | Mékrou Basin |
|---|---|---|---|---|
|  | count | count | count | count |
| Impacts of floods |  |  |  |  |
| Damage to the house | 3 | 24 | 5 | 32 |
| Loss of agricultural production | 8 | 44 | 7 | 59 |
| Injury or death of a household member |  | 1 | 3 | 4 |
| Loss of livestock |  | 16 | 4 | 20 |
| Impacts of droughts |  |  |  |  |
| Loss of agricultural production | 74 | 66 | 3 | 143 |
| Malnutrition of at least one household person | 41 | 4 | 2 | 47 |
| Loss of livestock | 20 | 4 | 3 | 27 |

*Table 679. Estimated costs of the recent flood and droughts (in FCFA)*

| | **Mékrou Basin** | **Benin** | **Burkina Faso** | **Niger** |
|---|---|---|---|---|
| | *Costs of floods* | | | |
| *Mean* | 324563 (~495 EURO) | 40.000 | 334.326 | 160.000 |
| *Standard Deviation* | 373159 | | 378.071 | |
| | *Costs of droughts* | | | |
| *Mean* | 256440 (~391 EURO) | 262.184 | 252.803 | 0 |
| *Standard Deviation* | 324224 | 257.676 | 362.150 | 0 |

Table 7810. Selected independent variables for modelling costs of floods and droughts

| Variable | Scaling | Correlation |
|---|---|---|
| *ECONSTAT* | 1. Rich, 2. Relatively rich, 3. Average, 4. Below average, 5. Much worse than average | CropProdLoss: -0.0786 LivestockLoss: -0.134 |
| *CropProdLoss* | 0 (no impact), 1 (impact) | LivestockLoss: -0.017 |
| *LivestockLoss* | 0 (no impact), 1 (impact) | PRODCROP: 0.212 |
| *PRODCROP* | Numerical Value | |

Table 8911. Costs of floods – Multivariate linear regression model

| **Number of obs = 48** | **R-sq: 0.4828** | **F-test: 7.842** | |
|---|---|---|---|
| **Independent Variables** | Coef. | Std. Err. | P>z |
| **ECONSTAT** | -128.548.5 | 47850.09 | 0.01 |
| **CropProdLoss** | 287.269.9 | 129224.5 | 0.031 |
| **LivestockLoss** | 359.272.4 | 92753.99 | 0 |
| **cons** | 429.595.2 | 225058.3 | 0.063 |
| **Cost Estimate / household** | 256.441 (FCFA) | 390.92 (EURO) | |

Table 9102. Costs of droughts - Multivariate linear regression model

| Number of obs = 98 | R-sq: 0.1907 | F-test: 11.196 | |
|---|---|---|---|
| **Independent Variables** | Coef. | Std. Err. | P>z |
| **PRODCROP** | 14.0 | 3.64 | 0 |
| **LivestockLoss** | 137,261.8 | 74243.4 | 0.068 |
| **_cons** | 130,367.5 | 39952.2 | 0.002 |
| **Cost Estimate / household** | 324,563 (FCFA) | 495 (EURO) | |

Figure 1. The Mékrou River Basin and the household survey area

[Figure]

[Figure]

Figure 2. Box-whiskers plot comparison of monthly precipitation for the first 25 years of the time series versus the last 10 years and Seasonal Mann-Kendall results. The Seasonal Mann-Kendal test performed did not highlight significant (p-value <0.010) increasing or decreasing trends in the analysis of the two precipitation time series.

[Figure]

[Figure]

2007-2016 vs. 1981-2006

**Last 10 years (2007-2016)**

| id | country | region | Mann-Kendall trend test for monthly environmental time series by Regions | | | | Seasonal Sen's slope and intercept | |
|---|---|---|---|---|---|---|---|---|
| | | | Kendall Score (S) | variance of S | two-sided p-value | Kendall's tau statistic | value of Sen's slope | intercept |
| 1 | Benin | Banikoara | -49 | 1491.6670 | 0.2045 | -0.0913 | -0.00250 | 35.238 |
| 2 | Benin | Pehunco | -42 | 1500.0000 | 0.2782 | -0.0778 | -0.06025 | 68.334 |
| 3 | Benin | Karimama | -21 | 1456.3330 | 0.5821 | -0.0400 | 0.00000 | 20.684 |
| 4 | Benin | Kounde | -36 | 1500.0000 | 0.3526 | -0.0667 | -0.04139 | 65.821 |
| 5 | Benin | Kerou | -40 | 1498.0000 | 0.3014 | -0.0742 | -0.02008 | 42.826 |
| 6 | Burkina Faso | Tansarga | -10 | 1389.3330 | 0.7885 | -0.0197 | 0.00000 | 24.686 |
| 7 | Burkina Faso | Bottou | 46 | 1371.3330 | 0.2142 | 0.0915 | 0.00000 | 16.893 |
| 8 | Burkina Faso | Diapaga | 18 | 1456.0000 | 0.6371 | 0.0343 | 0.00000 | 23.632 |
| 9 | Niger | Parc W | 43 | 1379.6670 | 0.2470 | 0.0852 | 0.00000 | 16.875 |
| 10 | Niger | Kirtachi | 26 | 1229.3330 | 0.4584 | 0.0587 | 0.00000 | 8.492 |
| 11 | Niger | Tamou | 66 | 1355.3330 | 0.0730 | 0.1326 | 0.00000 | 10.338 |
| 12 | Niger | Boboye | 12 | 1419.3330 | 0.7501 | 0.0232 | 0.00000 | 9.178 |

**All Time Series (1981-2016)**

| id | country | region | Mann-Kendall trend test for monthly environmental time series by Regions | | | | Seasonal Sen's slope and intercept | |
|---|---|---|---|---|---|---|---|---|
| | | | Kendall Score (S) | variance of S | two-sided p-value | Kendall's tau statistic | value of Sen's slope | intercept |
| 1 | Benin | Banikoara | 259 | 64624.3300 | 0.3083 | 0.0344 | 0.00052 | 31.010 |
| 2 | Benin | Pehunco | 349 | 64679.0000 | 0.1700 | 0.0462 | 0.01184 | 57.116 |
| 3 | Benin | Karimama | -74 | 62939.3300 | 0.7680 | -0.0100 | 0.00000 | 18.261 |
| 4 | Benin | Kounde | 421 | 64679.0000 | 0.0978 | 0.0557 | 0.01200 | 58.271 |
| 5 | Benin | Kerou | 397 | 64666.3300 | 0.1185 | 0.0526 | 0.00518 | 40.286 |
| 6 | Burkina Faso | Tansarga | 321 | 61339.6700 | 0.1949 | 0.0444 | 0.00000 | 24.718 |
| 7 | Burkina Faso | Bottou | 78 | 61251.3300 | 0.7526 | 0.0108 | 0.00000 | 13.290 |
| 8 | Burkina Faso | Diapaga | 324 | 62905.3300 | 0.1964 | 0.0440 | 0.00000 | 20.995 |
| 9 | Niger | Parc W | 112 | 59376.0000 | 0.6458 | 0.0160 | 0.00000 | 12.023 |
| 10 | Niger | Kirtachi | -149 | 56234.3300 | 0.5298 | -0.0225 | 0.00000 | 6.679 |
| 11 | Niger | Tamou | 79 | 59789.0000 | 0.7466 | 0.0112 | 0.00000 | 8.413 |
| 12 | Niger | Boboye | -132 | 61726.0000 | 0.5952 | -0.0182 | 0.00000 | 7.091 |

Figure 3. Temporal monthly precipitation profile for the Mékrou Area of Interest

[Figure]

[Figure]

Figure 4. HWMI index computed for 1981 to 2015 on the Mékrou

[Figure]

[Figure]

Figure 5. Mékrou River daily average discharge at Barou station as modelled in SWAT in the period 1995-2012. Dashed line indicates the average of the total period under consideration.

[Figure]

**APPENDIX A.**

QUESTIONNAIRE